# Beyond Real: Imaginary Extension of Rotary Position Embeddings for Long-Context LLMs

**Xiaoran Liu[1,2]\*, Yuerong Song[1,2]\*, Zhigeng Liu[1,2], Zengfeng Huang[1,2],**
**Qipeng Guo[2,3], Zhaoxiang Liu[4], Shiguo Lian[4], Ziwei He[2†], Xipeng Qiu[1,2†]**
[1]Fudan University, [2]Shanghai Innovation Institute, [3]Shanghai AI Lab, [4]China Unicom
`xrliu24@m.fudan.edu.cn, ziwei.he@sii.edu.cn, xpqiu@fudan.edu.cn`

## Abstract

Rotary Position Embeddings (RoPE) have become a standard for encoding sequence order in Large Language Models (LLMs) by applying rotations to query and key vectors in the complex plane. Standard implementations, however, utilize only the real component of the complex-valued dot product for attention score calculation. This simplification discards the imaginary component, which contains valuable phase information, leading to a potential loss of relational details crucial for modeling long-context dependencies. In this paper, we propose an extension that re-incorporates this discarded imaginary component. Our method leverages the full complex-valued representation to create a dual-component attention score. We theoretically and empirically demonstrate that this approach enhances the modeling of long-context dependencies by preserving more positional information. Furthermore, evaluations on a suite of long-context language modeling benchmarks show that our method consistently improves performance over the standard RoPE, with the benefits becoming more significant as context length increases. The code is available at `https://github.com/OpenMOSS/rope_pp`.

## 1 Introduction

Large Language Model (LLM) based on attention mechanism (Vaswani et al., 2017) now dominates Natural Language Processing (NLP) (OpenAI, 2023; Sun et al., 2024; OpenAI, 2024; Yang et al., 2025a), particularly in the long-context arena (Hassabis & Kavukcuoglu, 2024; Young et al., 2024; **?**), where attention overcomes the long-dependency bottlenecks of earlier architectures (LeCun et al., 1995; Schmidhuber et al., 1997). Recent work extends their context length to the million-token scale (Liu et al., 2024b; InternLM, 2025), and the key driver is position-embedding design (Su et al., 2024; Press et al., 2022; Peng et al., 2024). Among current LLMs, Rotary Position Embedding (RoPE) (Su et al., 2024) has become the canonical choice (Dubey et al., 2024; Meta, 2024a;b). It encodes the absolute position of every query and key vector $q_t, k_s$, namely token indices $s, t$ with a rotary matrix or complex multiplication, and when the two vectors make a dot product, it injects their relative position $t - s$, namely the relative distance, into the attention scores, thus combining the merits of traditional absolute and relative position embeddings (Vaswani et al., 2017; Dai et al., 2019; Yan et al., 2019) and securing widespread adoption.

Nevertheless, RoPE also has notable shortcomings, including poor length extrapolation (Press et al., 2022; Chen et al., 2023; bloc97, 2023), lack of data-sensitivity (Golovneva et al., 2024; Yang et al., 2025b), and no design for heterogeneous multi-modal input (Su, 2024a), prompting extensive research into its improvement. Most efforts concentrate on refining RoPE through interpolation designs (Peng et al., 2024; Liu et al., 2024d; Su, 2023), data-awareness (Zheng et al., 2024a;b), and feature-dimension partitioning (Wang et al., 2024; Wei et al., 2025). However, few work revisits the intrinsic computation of RoPE or analyze its inherent limitations (Hua et al., 2024; Dai et al., 2025). Re-examining RoPE in its complex-multiplication form reveals that the standard implementation keeps only the real part of the resulting complex attention score and discards the imaginary part outright (Su et al., 2024). Although taking the real part preserves the direct equivalence between complex multiplication and vector rotation, it incurs an irreversible information loss.

---

\* Equal contribution.    † Corresponding Author.

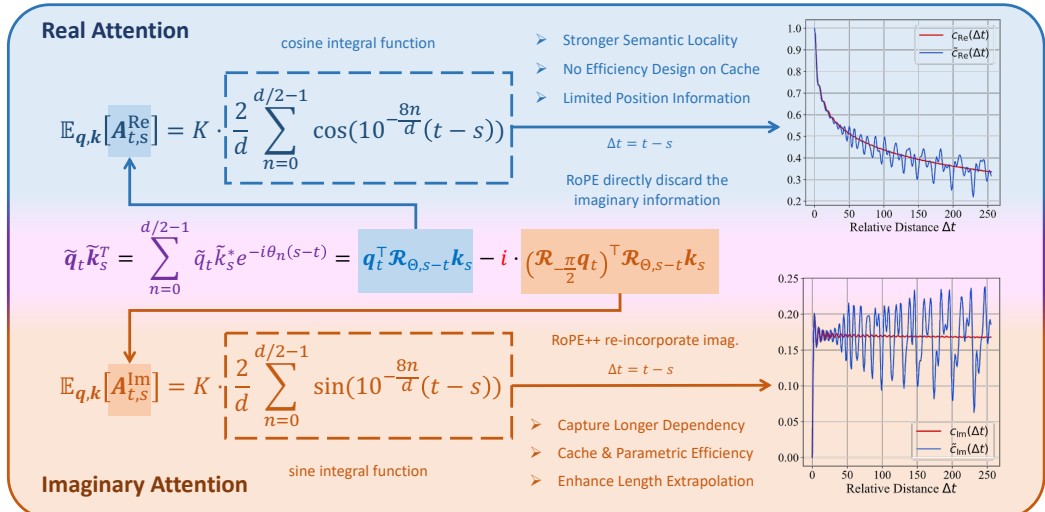

Figure 1: Overview of RoPE++. RoPE retains only the real part of the complex-valued attention score, whereas RoPE++ exploits the full complex representation to produce both real and imaginary attention. The real attention exhibits stronger semantic locality, while the imaginary attention preferentially captures long-context dependencies. RoPE++ combines the two, yielding multiple advantages.

A closer look at the imaginary attention, strictly, the negative imaginary part of attention, shows that, compared with the real attention exhibiting stronger semantic locality, the imaginary heads attend more to long-context information as shown in Figure 1, promising gains on long-context tasks. Moreover, adding imaginary attention also exposes $q_t, k_s$ to a wider positional information range, implicitly improving length extrapolation. Therefore, we propose **RoPE++**, as illustrated in Figure 1, which re-injects the discarded imaginary component as a new group of attention heads computed in parallel with the real attentions. Particularly, we introduce **RoPE++$_{EH}$** that keeps equal attention head number while halving QKV parameters as well as KV cache, and **RoPE++$_{EC}$** that keeps equal cache size and doubles the number of attention heads. Theoretical analysis and pre-training experiments validate the above advantages. Both RoPE++$_{EH}$ and RoPE++$_{EC}$ outperform vanilla RoPE and other position embeddings on general tasks. On long-context benchmarks, RoPE++$_{EH}$ achieves comparable results with vanilla RoPE with half the cache, whereas RoPE++$_{EC}$ outperforms significantly at the same cache cost. Our contributions can be summarized as follows:

- We first identify the loss of imaginary information in standard RoPE and find it advantageous for capturing long-context dependencies by analyzing the properties of imaginary attention.

- Building on this, we propose RoPE++, which reintroduces the imaginary computation into attention in two configurations, RoPE++$_{EH}$ with equal head number and halved KV cache, and RoPE++$_{EC}$ with equal cache size and doubled attention heads. Both preserve the unified absolute–relative position-embedding format.

- Pre-training and evaluation at 376M and 776M sizes show that RoPE++$_{EH}$ and RoPE++$_{EC}$ outperform vanilla RoPE and other position embeddings on average across short- and long-context benchmarks. Further analysis reveals that the imaginary attentions play a dominant role in modeling long-context dependencies, confirming the effectiveness of introducing imaginary attention for improved long-context capability.

## 2 RELATED WORK

Rotary Position Embedding (RoPE) is the dominant position embedding in current LLMs (Dubey et al., 2024; Meta, 2024a;b; Yang et al., 2025a). We analyze its good properties in Appendix B, including unifying relative and absolute information via rotation matrices and complex multiplication, and semantic aggregation as well as long-context decay. Yet it still faces many other challenges,

attracting a great deal of effort to its improvement as mentioned above. A large body of work targets length extrapolation, scaling the rotary base (bloc97, 2023; Liu et al., 2024d; Xiong et al., 2024), interpolating or compressing index ranges (Press et al., 2022; Peng et al., 2024; Jin et al., 2024), or coupling RoPE with sparse attention (Lu et al., 2024; Xiao et al., 2024a; Liu et al., 2024c) to let models process contexts far longer than the training window. Other efforts extend RoPE to heterogeneous, cross-modal inputs (Su, 2024a), especially text–video sequences (Wang et al., 2024; Wei et al., 2025). Parallel lines design parametric schemes that encode contextual cues (Golovneva et al., 2024; Zheng et al., 2024a; Lin et al., 2025), refining or replacing RoPE to yield data-dependency.

However, few works revisit RoPE's intrinsic computation or analyze its inherent limitations (Hua et al., 2024; Yang et al., 2025b; Dai et al., 2025). Particularly, the imaginary information loss of RoPE in rotation format compared with the complex multiplication format remains overlooked. Although prior work has tried to incorporate the full complex computation into the self-attention mechanism or neural networks (Wang et al., 2025; Lee et al., 2022), the characteristics and functionality of the imaginary component in position embedding remain unexplored. Therefore, we propose RoPE++ and close this gap through a deep analysis of the mathematical properties of imaginary attention and extensive validation on both short- and long-context downstream tasks.

## 3 METHODOLOGY

We begin our method by revisiting the complex form of RoPE. Only the real part of the complex product is retained, and the imaginary part is discarded, as shown in Equation 1. Although current LLMs perform well with this real-only attention, omitting the imaginary component may remove physical information. LLM no longer sees the full magnitude and phase of the complex attention result. This raises the question: can the imaginary part be re-incorporated into the attention computation?

$$
\begin{aligned}
\boldsymbol{A}_{t,s} &= \mathrm{Re}\left[\sum_{n=0}^{d/2-1} \tilde{q}_t^{(n)} \tilde{k}_s^{(n)*} e^{-i\theta_n(t-s)}\right] = \mathrm{Re}\left[\sum_{n=0}^{d/2-1} \left(\tilde{q}_t^{(n)} e^{-i\theta_n t}\right)\left(\tilde{k}_s^{(n)} e^{-i\theta_n s}\right)^*\right] \\
&= \sum_{n=0}^{d/2-1} \left(q_t^{(2n)} k_s^{(2n)} + q_t^{(2n+1)} k_s^{(2n+1)}\right) \cos\theta_n(t-s) + \\
&\quad\ \left(q_t^{(2n)} k_s^{(2n+1)} - q_t^{(2n+1)} k_s^{(2n)}\right) \sin\theta_n(t-s)
\end{aligned}
\tag{1}
$$

In this section, we will first propose our RoPE++ by re-introducing the imaginary information, in Section 3.1, as a new group of attention heads, namely imaginary attentions, compared with original real attentions. We then analyze the strengths from three aspects, the imaginary heads' stronger capture of long-context dependencies in Section 3.2, the cache and parameter reduction by combining imaginary and real heads in Section 3.3, and the impact on length extrapolation in Section 3.4.

### 3.1 IMAGINARY EXTENSION OF ROPE

We first recover the imaginary part that is discarded in Equation 1. The resulting expression is given in Equation 2. Strictly speaking, it is the negative imaginary part, and the reason will be detailed in Section 3.2. Similar to the real part, the imaginary part carries relative position information between $\boldsymbol{q}_t, \boldsymbol{k}_s$, so the formula can be rearranged into a vector form as shown in Equation 2.

$$
\begin{aligned}
\boldsymbol{A}_{t,s}^{\mathrm{Im}} &= -\mathrm{Im}\left[\sum_{n=0}^{d/2-1} \tilde{q}_t^{(n)} \tilde{k}_s^{(n)*} e^{-i\theta_n(t-s)}\right] = -\mathrm{Im}\left[\sum_{n=0}^{d/2-1} \left(\tilde{q}_t^{(n)} e^{-i\theta_n t}\right)\left(\tilde{k}_s^{(n)} e^{-i\theta_n s}\right)^*\right] \\
&= \sum_{n=0}^{d/2-1} \left(q_t^{(2n)} k_s^{(2n)} + q_t^{(2n+1)} k_s^{(2n+1)}\right) \sin\theta_n(t-s) - \\
&\quad\ \left(q_t^{(2n)} k_s^{(2n+1)} - q_t^{(2n+1)} k_s^{(2n)}\right) \cos\theta_n(t-s)
\end{aligned}
\tag{2}
$$

We observe that the imaginary attention still follows a rotation form and can be decomposed into absolute position embeddings on $\boldsymbol{q}_t, \boldsymbol{k}_s$, as shown in Equation 3. Specifically, the embedding applied

to $\boldsymbol{k}_s$ is identical to that used in the real attention in Equation 6 in Appendix B. For $\boldsymbol{q}_t$, the embedding is equivalent to rotating the vector by $-\pi/2$ before applying the same embedding in the real case.

$$
\boldsymbol{A}_{t,s}^{\mathrm{Im}} = \underbrace{\sum_{n=0}^{d/2-1} \begin{bmatrix} q_t^{(2n+1)} \\ -q_t^{(2n)} \end{bmatrix}^\top \begin{bmatrix} \cos\theta_n(t-s) & \sin\theta_n(t-s) \\ -\sin\theta_n(t-s) & \cos\theta_n(t-s) \end{bmatrix} \begin{bmatrix} k_s^{(2n)} \\ k_s^{(2n+1)} \end{bmatrix}}_{\text{Relative PE}}
$$

$$
= \underbrace{\sum_{n=0}^{d/2-1} \left( \begin{bmatrix} \cos\theta_n t & -\sin\theta_n t \\ \sin\theta_n t & \cos\theta_n t \end{bmatrix} \begin{bmatrix} q_t^{(2n+1)} \\ -q_t^{(2n)} \end{bmatrix} \right)^\top \left( \begin{bmatrix} \cos\theta_n s & -\sin\theta_n s \\ \sin\theta_n s & \cos\theta_n s \end{bmatrix} \begin{bmatrix} k_s^{(2n)} \\ k_s^{(2n+1)} \end{bmatrix} \right)}_{\text{Absolute PE}}
$$

$(3)$

We thus obtain an expression for the imaginary attention, strictly speaking, the negative imaginary attention. If we denote the rotation matrix as $\boldsymbol{\mathcal{R}}_{.}$ and $\boldsymbol{\mathcal{R}}_{\Theta,.}$. The latter is parameterized with $\theta_0, \cdots, \theta_{d/2-1}$. The computation of real and imaginary attention can be summarized in Equation 4.

$$
\boldsymbol{A}_{t,s}^{\mathrm{Re}} = \mathrm{Re} \left[ \sum_{n=0}^{d/2-1} \tilde{q}_t^{(n)} \tilde{k}_s^{(n)*} e^{i\theta_n(s-t)} \right] = (\boldsymbol{\mathcal{R}}_{\Theta,t} \boldsymbol{q}_t)^\top \boldsymbol{\mathcal{R}}_{\Theta,s} \boldsymbol{k}_s = \boldsymbol{q}_t^\top \boldsymbol{\mathcal{R}}_{\Theta,s-t} \boldsymbol{k}_s
$$

$$
\boldsymbol{A}_{t,s}^{\mathrm{Im}} = -\mathrm{Im} \left[ \sum_{n=0}^{d/2-1} \tilde{q}_t^{(n)} \tilde{k}_s^{(n)*} e^{i\theta_n(s-t)} \right] = \left( \boldsymbol{\mathcal{R}}_{\Theta,t} \boldsymbol{\mathcal{R}}_{-\frac{\pi}{2}} \boldsymbol{q}_t \right)^\top \boldsymbol{\mathcal{R}}_{\Theta,s} \boldsymbol{k}_s = (\boldsymbol{\mathcal{R}}_{-\frac{\pi}{2}} \boldsymbol{q}_t)^\top \boldsymbol{\mathcal{R}}_{\Theta,s-t} \boldsymbol{k}_s
$$

$(4)$

Notably, the newly introduced imaginary component retains the key property of the original RoPE, that it can still be formulated either as a relative position or as an absolute position embedding. The only required adjustment is to rotate $\boldsymbol{q}_t$ by $-\pi/2$ and then apply the standard position embedding to obtain the imaginary term. We refer to RoPE augmented with this imaginary extension as **RoPE++**. This augmentation raises further questions: what semantics does the imaginary attention convey, does it introduce additional overhead, and can it enhance model performance?

### 3.2 CAPTURE LONGER DEPENDENCY

As stated in Preliminary in Appendix B, the original RoPE-based attention or real attention exhibits *semantic aggregation* and *long-context decay*, both governed by its characteristic curve, as shown in Equation 7 and Figure 1. Similarly, we can derive the characteristic curve for the imaginary attention in RoPE++. It is the average of $\sin(\theta\Delta t)$ over the same frequency distribution, approximating a sine integral function as shown in Equation 5 and Figure 1.

$$
c_{\mathrm{Im}}(\Delta t) = \frac{2}{d} \sum_{n=0}^{d/2-1} \sin\left( 10^{-\frac{8n}{d}} \Delta t \right), \quad \tilde{c}_{\mathrm{Im}} = \int_{10^{-4}}^{1} \frac{\sin\theta t}{\theta \ln 10^4} \mathrm{d}\theta = \mathrm{Si}(\Delta t) - \mathrm{Si}\left( \frac{\Delta t}{10^4} \right) \quad (5)
$$

Although modeling distance with $\sin(\theta\Delta t)$ is counter-intuitive, since $\sin(\theta\Delta t)$ is zero at zero relative distance, rises, then falls, unlike $\cos(\theta\Delta t)$'s monotonic drop in the first half-period, the characteristic curve of the imaginary attention still shares the semantic-aggregation property of the real part. For $\Delta t > 0$, when $\boldsymbol{q}_t, \boldsymbol{k}_s$ are similar, their attention is on average larger regardless of relative distance, which is the reason why we take the negative imaginary part as imaginary attention. Moreover, on average, this component attends more to distant positions. As shown in Figure 1, its characteristic curve declines very slowly beyond a certain distance. Consequently, the imaginary part assigns more weight to the long-context region than the real part, helping LLM retrieve long-context information.

### 3.3 CACHE AND PARAMETRIC EFFICIENCY

As described earlier, computing the imaginary attention requires only rotating the $\boldsymbol{q}_t$ by $-\pi/2$, while every other operation is identical to the original RoPE. Because the positional embedding of $\boldsymbol{k}_s$ is unchanged, we can interleave the $-\pi/2$-rotated $\boldsymbol{q}_t$ with the original $\boldsymbol{q}_t$ and perform the real and imaginary attention in a single pass in FlashAttention (Dao, 2024). Consequently, no extra KV

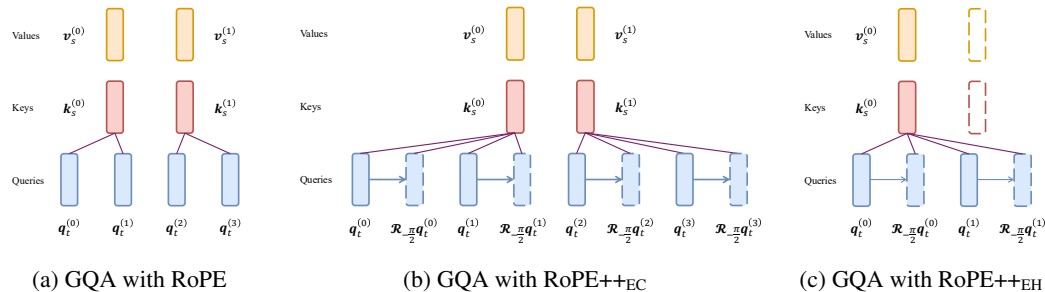

(a) GQA with RoPE      (b) GQA with RoPE++$_{\text{EC}}$      (c) GQA with RoPE++$_{\text{EH}}$

Figure 2: Visualization of GQA with different RoPE schema. RoPE++$_{\text{EC}}$ shares equal cache and twice the attention head with RoPE, while RoPE++$_{\text{EH}}$ has equal attention head and half the KV cache.

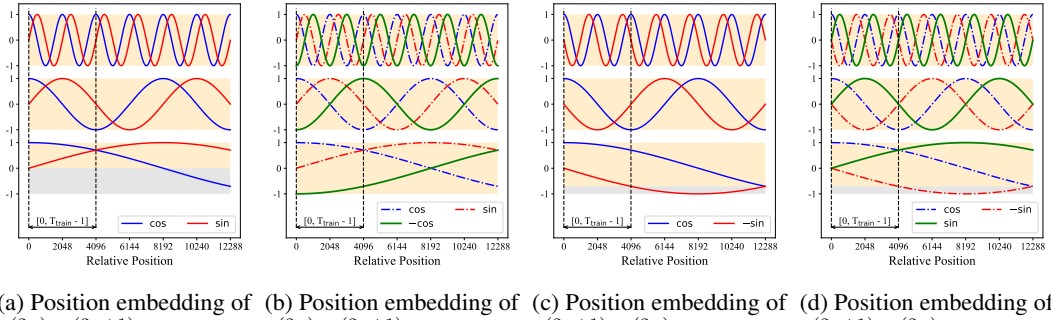

(a) Position embedding of $q^{(2n)}, k^{(2n+1)}$ in RoPE

(b) Position embedding of $q^{(2n)}, k^{(2n+1)}$ in RoPE++

(c) Position embedding of $q^{(2n+1)}, k^{(2n)}$ in RoPE

(d) Position embedding of $q^{(2n+1)}, k^{(2n)}$ in RoPE++

Figure 3: Comparison of trained position embedding interval between RoPE and RoPE++. The area within the dashed line represents trained relative position, and that beyond is in length extrapolation, with learned position embedding values colored in yellow and the opposite in gray.

cache is introduced, and the method plugs directly into MHA or GQA (Ainslie et al., 2023), merely doubling the attention head group size, as shown in Figure 2b. We refer to this configuration as **RoPE++$_{\text{EC}}$**, namely RoPE++ with equal cache size. The only cost of RoPE++$_{\text{EC}}$ is an additional imaginary attention computed alongside the real one under the fixed QKV parameter budget.

Conversely, if the total head number is kept fixed, both QKV parameters and KV cache sizes are halved. We refer to this configuration as **RoPE++$_{\text{EH}}$**, namely RoPE++ with equal attention head number, as shown in Figure 2c. In long-context scenarios, RoPE++$_{\text{EH}}$ halves the cache and raises throughput. Because the imaginary attention doubles the number of output heads, $\boldsymbol{W}_o$ must be twice as large as $\boldsymbol{W}_q$. Therefore, $\boldsymbol{W}_o$ in RoPE++$_{\text{EH}}$ equals the original RoPE size, whereas $\boldsymbol{W}_o$ in RoPE++$_{\text{EC}}$ is double-sized. Experiments in Section 4 show that RoPE++$_{\text{EC}}$ outperforms the original RoPE, especially on long-context tasks, and RoPE++$_{\text{EH}}$ delivers comparable or even superior results.

Importantly, the imaginary and real attention, though computed independently and treated as separate heads, must share the same parameter. Both RoPE++$_{\text{EH}}$ and RoPE++$_{\text{EC}}$ share $\boldsymbol{W}_q$ between the real and imaginary attention. Allocating distinct subsets of heads to imaginary and real attention would effectively collapse back to standard RoPE, since rotating $\boldsymbol{q}_t$ in imaginary attention by $\pi/2$ yields real attention, with no architecture modification. In other words, imaginary attention is defined relative to real attention and cannot exist independently. Therefore, configurations such as 75% imaginary vs. 25% real or 100% imaginary (applying only the imaginary part) are impossible under RoPE++.

### 3.4 IMPACT ON LENGTH EXTRAPOLATION

A closer inspection of the real and imaginary attention computations reveals an interesting discovery. In vanilla RoPE-based attention, or real attention, as shown in Equation 6, even-index query dimensions $\boldsymbol{q}^{(2n)}$ and odd-index key dimensions are multiplied only by $\cos\theta_n(t-s)$ and $\sin\theta_n(t-s)$

whose values are always non-negative when $\theta_n$ is small. Once the input length exceeds the pre-training context length, these dimensions encounter out-of-distribution (OOD) negative embeddings as shown in Figure 5f and thus extrapolate poorly (Liu et al., 2024d; Peng et al., 2024). In RoPE++ as shown in Equation 3, these dimensions are multiplied by $-\cos\theta_n(t-s)$ and $\sin\theta_n(t-s)$ in the imaginary attention, so during pre-training, they have already observed both negative and positive position embedding as well as their maximum and minimum value $\pm 1$. Consequently, these dimensions no longer suffer from the length extrapolation problem in longer contexts (Liu et al., 2025b).

Likewise, odd-index query dimensions $\boldsymbol{q}^{(2n+1)}$ and even-index key dimensions $\boldsymbol{k}^{(2n)}$ encounter only $\cos\theta_n(t-s)$ and $-\sin\theta_n(t-s)$ in the real attention, and the imaginary attention further exposes them to $\cos\theta_n(t-s)$ and $\sin\theta_n(t-s)$. Yet this alone does not expand the position embedding range trained in pre-training, as shown in Figure 5h and Figure 5j. However, when real and imaginary attention are combined, $\boldsymbol{q}_t, \boldsymbol{k}_s$ in RoPE++ attains the full $\cos$ and $\sin$ value range, once the training length exceeds half the sinusoidal period, whereas the vanilla RoPE requires a full period. Consequently, more dimensions in RoPE++ observe complete positional information. Therefore, perplexity grows more slowly beyond the maximum supported context length (Liu et al., 2024d; Men et al., 2024).

## 4 EXPERIMENT

### 4.1 SETUP

We validate RoPE++ at both 776M and 376M model sizes, with architectural details in Appendix C. Both models are pre-trained on DCLM-Baseline-1.0 corpus (Li et al., 2024) by HuggingFace Transformers (Wolf et al., 2020) on 8 NVIDIA H200 160 GB GPUs. For each size, we use a batch size of 0.5M tokens and pre-train for 50B tokens. We use AdamW (Loshchilov et al., 2017) optimizer with weight decay 0.1, a maximum learning rate of 5e-4, and a warmup-stable-decay scheduler. We use the first 0.5B tokens for warmup, and the final 5B tokens for decay, and the learning rate ends at 0.

We compare our RoPE++ with standard RoPE (Su et al., 2024) and other well-known position embedding designs, including FoPE (Hua et al., 2024), Pythia (namely, partial RoPE with only last 1/4 dimensions being rotated) (Biderman et al., 2023), as well as ALiBi (Press et al., 2022). We pre-train all methods on 4k context length with an initial rotary base of 10000. For RoPE and RoPE++, we conduct continuous long-context pre-training. Following Xiong et al. (2024); Lv et al. (2024), we scale the rotary base from 10000 to 500000 and train for 10B tokens from DCLM on 32k context length, using a cosine-annealing learning rate scheduler and keeping all other settings.

### 4.2 SHORT-CONTEXT EVALUATION

We evaluate both short-context and long-context tasks based on OpenCompass (Contributors, 2023). For short-context evaluation, we measure perplexity on WikiText (Merity et al., 2017) and LAMBADA(Paperno et al., 2016) and assess downstream tasks mainly in Open LLM Leaderboard (HuggingFace, 2023), including TruthfulQA (Lin et al., 2022), PIQA (Bisk et al., 2020), HellaSwag (Zellers et al., 2019), WinoGrande (Sakaguchi et al., 2020), ARC-e (Clark et al., 2018), GPQA (Rein et al., 2023), SocialIQA (Sap et al., 2019), OpenBookQA (Mihaylov et al., 2018), and SuperGLUE (Wang et al., 2019). All models are tested within a 4k context length.

The results are shown in Table 1. Our RoPE++$_{EC}$ and RoPE++$_{EH}$ achieve the best average scores on short-context tasks compared with RoPE and every other position embedding design. Notably, RoPE++$_{EH}$ surpasses standard RoPE with only half the KV-cache and QKV parameters. After further long-context pre-training, RoPE++ still retains this edge over RoPE on short-text benchmarks.

### 4.3 LONG-CONTEXT EVALUATION

For long-context evaluation, we evaluate downstream performance at varying lengths with the classical synthetic benchmarks, RULER (Hsieh et al., 2024) and BABILong (Kuratov et al., 2024). The results are shown in Table 2 and Figure 6. We highlight the comparison with RoPE in long-context training because RoPE is the position embedding currently most widely used by long-context LLMs.

On RULER and BABILong up to 64k context, our RoPE++ again acquires the highest scores. Particularly, RoPE++$_{EH}$ achieves comparable performance with vanilla RoPE using half the KV-

| | Wiki | LMB | TQA | PIQA | Hella | Wino | ARC-e | GPQA | SIQA | OBQA | SG | Avg. |
|---|---|---|---|---|---|---|---|---|---|---|---|---|
| | ppl ↓ | ppl ↓ | acc ↑ | acc ↑ | acc ↑ | acc ↑ | acc ↑ | acc ↑ | acc ↑ | acc ↑ | acc ↑ | |
| ***376M Short*** | | | | | | | | | | | | |
| RoPE | 19.9 | 32.7 | 35.5 | 66.3 | 34.8 | 50.9 | 39.3 | 24.8 | 38.6 | **27.4** | 43.7 | 40.1 |
| FoPE | 19.3 | 33.0 | 33.8 | 65.9 | 34.5 | **53.0** | 37.0 | **28.8** | 39.5 | 24.2 | 43.6 | 40.0 |
| Pythia | **19.2** | 32.9 | 34.7 | 65.8 | 34.9 | 51.5 | 41.3 | 21.2 | 39.7 | 25.6 | 42.5 | 39.7 |
| ALiBi | 21.2 | 34.6 | 33.8 | 66.1 | 34.2 | 51.1 | **44.4** | 24.8 | 38.7 | 27.4 | 43.9 | 40.5 |
| RoPE++$_{EH}$ | 20.8 | 33.6 | 36.3 | 66.4 | 34.5 | 52.5 | 40.9 | 23.7 | **40.5** | 24.8 | 43.2 | 40.3 |
| RoPE++$_{EC}$ | 19.4 | **32.6** | **37.3** | **68.0** | **35.6** | **53.0** | 41.3 | 25.8 | 40.3 | 23.2 | **44.8** | **41.0** |
| ***376M Long*** | | | | | | | | | | | | |
| RoPE | 20.4 | **33.8** | 35.4 | 64.9 | 34.1 | 50.6 | 40.4 | 21.2 | **39.4** | 27.4 | 43.5 | 39.6 |
| RoPE++$_{EH}$ | 21.7 | 34.8 | 35.2 | 64.5 | **34.3** | 49.9 | **41.5** | **22.7** | 40.0 | 27.0 | 43.1 | 39.8 |
| RoPE++$_{EC}$ | **20.0** | 33.9 | **37.1** | **66.1** | 34.1 | **53.4** | 38.1 | 21.2 | 39.2 | **28.4** | **43.7** | **40.1** |
| ***776M Short*** | | | | | | | | | | | | |
| RoPE | 14.8 | 27.3 | 35.5 | **70.1** | **43.7** | 52.3 | 43.4 | 25.8 | 41.3 | 21.8 | 43.6 | 42.0 |
| FoPE | **14.7** | 27.1 | 33.6 | 68.7 | 43.4 | 52.9 | **45.0** | 24.8 | 39.7 | 24.8 | 45.4 | 42.0 |
| Pythia | 14.8 | **26.9** | 35.8 | 68.8 | 42.9 | 52.1 | 39.5 | 22.2 | 42.0 | 21.2 | 43.6 | 40.9 |
| ALiBi | 15.2 | 28.3 | 35.2 | 70.2 | **43.7** | 53.6 | 43.2 | 23.7 | 40.6 | 27.6 | **45.9** | 42.6 |
| RoPE++$_{EH}$ | 15.6 | 28.1 | 35.4 | 69.6 | 42.7 | 53.5 | **45.0** | 15.8 | **41.6** | 26.8 | 42.4 | 42.5 |
| RoPE++$_{EC}$ | 14.8 | 27.3 | **36.1** | 69.3 | 43.6 | 52.3 | 43.7 | **28.3** | 40.1 | 27.6 | 44.4 | **42.8** |
| ***776M Long*** | | | | | | | | | | | | |
| RoPE | 14.6 | 27.3 | 35.1 | 68.9 | 43.1 | 51.5 | **47.6** | 21.7 | 40.7 | 20.2 | 42.6 | 41.3 |
| RoPE++$_{EH}$ | 15.3 | 28.1 | **35.4** | 69.9 | 41.9 | **52.6** | 43.2 | 28.3 | **41.0** | 22.2 | 43.4 | 42.0 |
| RoPE++$_{EC}$ | **14.4** | **27.1** | 35.2 | **70.4** | **43.7** | **52.6** | 44.8 | **31.8** | 40.8 | **27.6** | **44.3** | **43.5** |

Table 1: Results on short-context tasks for 776M and 376M models pre-trained in 4k context length and further trained on 32k. Best results are highlighted in bold, with the second best underlined for broader comparison. Our RoPE++ achieves the best average performance on different model sizes.

| | RULER | | | | | | BABILong | | | | | | |
|---|---|---|---|---|---|---|---|---|---|---|---|---|---|
| | 4k | 8k | 16k | 32k | 64k | Avg. | 2k | 4k | 8k | 16k | 32k | 64k | Avg. |
| ***376M Long*** | | | | | | | | | | | | | |
| RoPE | 31.6 | 25.6 | 22.0 | 9.5 | 5.5 | 18.8 | 17.7 | 16.1 | 9.1 | 9.4 | 5.9 | 7.8 | 11.0 |
| RoPE++$_{EH}$ | 29.9 | 28.4 | 17.6 | 9.4 | 5.9 | 18.2 | 14.1 | 15.6 | 12.2 | 9.9 | 8.3 | 9.7 | 11.6 |
| RoPE++$_{EC}$ | **36.1** | **33.0** | **29.1** | **17.7** | 9.0 | **25.0** | **19.8** | **19.8** | **16.1** | **15.8** | **12.3** | **12.8** | **16.1** |
| ***776M Long*** | | | | | | | | | | | | | |
| RoPE | 37.4 | 35.1 | 33.0 | 21.2 | 10.4 | 27.4 | **33.5** | **30.7** | 23.6 | 22.0 | 15.1 | 12.1 | 22.8 |
| RoPE++$_{EH}$ | 38.7 | 35.4 | **33.8** | 24.6 | 10.7 | 28.6 | 31.9 | 26.5 | 18.6 | 16.2 | 11.0 | 12.2 | 19.4 |
| RoPE++$_{EC}$ | **42.7** | **38.6** | 33.4 | 21.7 | **10.9** | **29.4** | 32.4 | 29.9 | **24.4** | **24.5** | **18.6** | **14.8** | **24.1** |

Table 2: Results on long-context tasks, including RULER and BABILong for 776M and 376M models further trained with 5B tokens in 32k context length. Best results are highlighted in bold. Our RoPE++ achieves the best performance on average, especially in long-context scenarios.

cache and QKV parameters, while RoPE++$_{EC}$ delivers significant gains at the same cache size. Although RoPE occasionally edges ahead at a few shorter context lengths, RoPE++, including both RoPE++$_{EC}$ and RoPE++$_{EH}$, maintains more stable performance as context length grows and achieves best performance in 64k context length extrapolation consistently.

## 5 DISCUSSION

### 5.1 ROPE++ AS CACHE OPTIMIZATION

As mentioned in Section 3.3, RoPE++$_{EH}$ halves KV cache and QKV parameters while keeping the attention head number equal, yielding evident efficiency gains. We validate this efficiency strength by assessing the memory cost as well as Time-Per-Output-Token (TPOT) of 376M and 776M models,

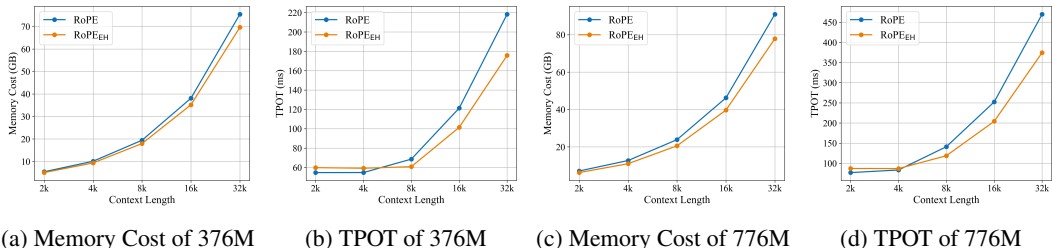

(a) Memory Cost of 376M  (b) TPOT of 376M  (c) Memory Cost of 776M  (d) TPOT of 776M

Figure 4: Efficiency comparison between RoPE and RoPE++$_{\text{EH}}$ in 376M and 776M model. RoPE++$_{\text{EH}}$ lowers memory cost and accelerates decoding, and the margin widens as context grows.

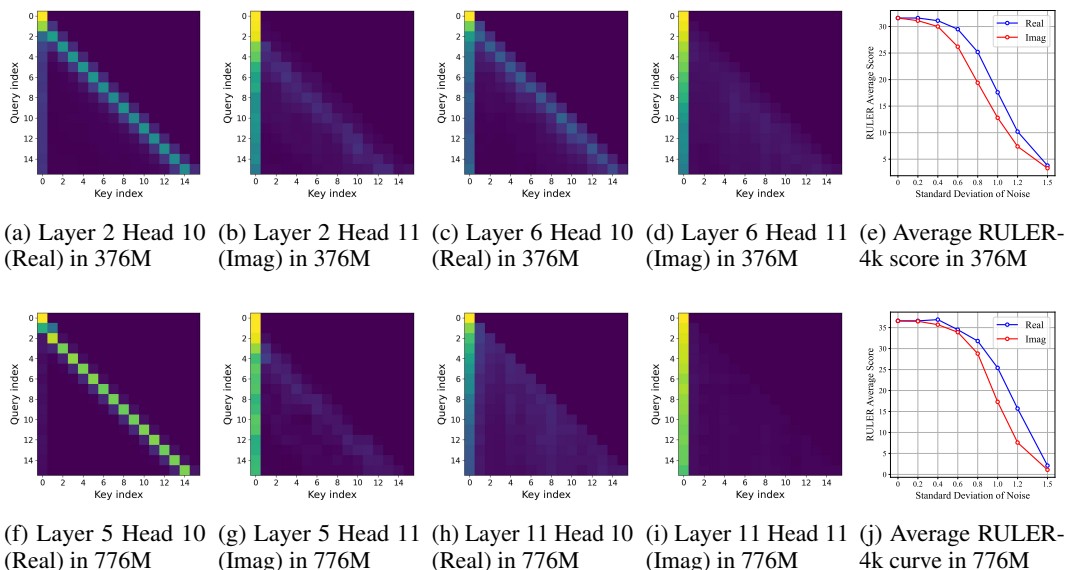

(a) Layer 2 Head 10 (Real) in 376M  (b) Layer 2 Head 11 (Imag) in 376M  (c) Layer 6 Head 10 (Real) in 376M  (d) Layer 6 Head 11 (Imag) in 376M  (e) Average RULER-4k score in 376M

(f) Layer 5 Head 10 (Real) in 776M  (g) Layer 5 Head 11 (Imag) in 776M  (h) Layer 11 Head 10 (Real) in 776M  (i) Layer 11 Head 11 (Imag) in 776M  (j) Average RULER-4k curve in 776M

Figure 5: Attention-score patterns and long-context performance in 376M and 776M RoPE++ models. Imaginary heads attend markedly to global information, whereas real heads focus more on local context. Adding Gaussian noise to imaginary attention degrades long-context performance more severely, over 8 points, than the same perturbation applied to real attention.

from 2k to 32k context length. We conduct the efficiency evaluation on a single NVIDIA H200 160BG GPU, with a batch size of 8 samples. The results are shown in Figure 4. At both 376M and 776M, RoPE++$_{\text{EH}}$ consistently reduces memory cost and speeds up decoding, with the margin widening as context length increases.

## 5.2 ATTENTION PATTERN OF RoPE++

To verify how imaginary attention captures long-context dependencies and to contrast it with real attention in RoPE++, we inspect the attention patterns of short-context-trained RoPE++$_{\text{EC}}$ at 376M and 776M as shown in Figure 5. Odd-index imaginary attention highlights the initial positions more strongly than even-index real heads, indicating a stronger global focus. Since prior work (Liu et al., 2025a; Wei et al., 2025) shows that dimensions attending globally are more critical for long-context semantics, imaginary attention may play the dominant role in long-context tasks.

For further verification, we design the following validation experiment. We add Gaussian noise with equal standard deviation to the imaginary and real attention components separately, and monitor the change in RoPE++ performance on long-context tasks, such as the average score of RULER-4k. Curves for RULER-4k versus standard deviation are plotted for both real and imaginary attention. When the standard deviation $\sigma$ is small ($\sigma < 0.2$), scores with corrupted real or imaginary attentions stay close to the baseline; when it is large enough ($\sigma = 1.5$), both drop sharply. Importantly, in the

| | **Short** | | **RULER** | | | | | **BABILong** | | | | | |
|---|---|---|---|---|---|---|---|---|---|---|---|---|---|
| | ppl | score | 4k | 8k | 16k | 32k | Avg | 2k | 4k | 8k | 16k | 32k | Avg |
| *376M Long PI* | | | | | | | | | | | | | |
| RoPE | **33.4** | 42.0 | 36.5 | **33.6** | 19.7 | **10.6** | 25.1 | 19.3 | 12.3 | 10.2 | 10.9 | 10.9 | 12.7 |
| RoPE++$_{EH}$ | 34.7 | 41.7 | 28.0 | 27.6 | 15.8 | 6.9 | 19.6 | 13.3 | 12.4 | 12.8 | 8.9 | 10.4 | 11.6 |
| RoPE++$_{EC}$ | 33.7 | **42.8** | **37.0** | 32.4 | **28.3** | **10.6** | **27.1** | **24.0** | **20.7** | **15.9** | **14.3** | **12.3** | **17.4** |
| *376M Long YaRN* | | | | | | | | | | | | | |
| RoPE | **32.8** | 42.2 | **36.4** | 32.9 | 28.4 | 15.0 | 28.2 | 22.4 | 16.4 | 11.4 | 10.7 | 11.1 | 14.4 |
| RoPE++$_{EH}$ | 33.9 | 42.2 | 32.7 | 30.2 | 24.9 | 10.7 | 24.7 | 8.7 | 9.3 | 12.1 | 11.3 | 10.9 | 10.5 |
| RoPE++$_{EC}$ | 32.9 | **43.4** | 36.0 | **33.9** | **31.7** | **17.8** | **29.8** | **27.4** | **23.6** | **18.0** | **16.9** | **12.3** | **19.6** |
| *776M Long PI* | | | | | | | | | | | | | |
| RoPE | **27.8** | 40.4 | 37.8 | 34.4 | **30.5** | 13.4 | 29.0 | 15.3 | 16.9 | 12.7 | 11.8 | 9.3 | 13.2 |
| RoPE++$_{EH}$ | 28.8 | 40.4 | 37.9 | 35.0 | 27.5 | **14.6** | 28.8 | 21.0 | 22.4 | **17.1** | **13.7** | **11.1** | **17.1** |
| RoPE++$_{EC}$ | **27.8** | **40.5** | **43.0** | **38.7** | 28.8 | 13.6 | **31.0** | **25.7** | **23.4** | 16.4 | 9.4 | 8.0 | 16.6 |
| *776M Long YaRN* | | | | | | | | | | | | | |
| RoPE | 27.3 | 40.9 | 37.6 | 35.0 | 33.9 | **27.5** | 33.5 | 26.9 | **25.6** | 19.5 | 16.4 | 12.2 | 20.1 |
| RoPE++$_{EH}$ | 28.3 | 40.6 | 37.9 | 34.9 | 32.2 | 26.1 | 32.8 | **28.0** | 23.9 | 18.6 | 17.8 | 11.7 | 20.0 |
| RoPE++$_{EC}$ | **27.3** | **41.5** | **42.9** | **36.5** | **36.3** | 22.2 | **34.4** | 26.3 | 24.1 | **21.1** | **19.8** | **16.9** | **21.6** |

Table 3: Results of 776M and 376M models further trained with 5B tokens in 32k context length with YaRN and Linear PI. Our RoPE++ still achieves the best performance on average.

intermediate range, adding noise to the imaginary attention always performs worse than corrupting the real part. When $\sigma = 1.0$, for example, the real-noised RoPE++ outperforms the imaginary-noised one by 5 points at 376M and 8 points at 776M, which demonstrates a significant gap. Thus, impairing the imaginary heads degrades long-context performance more, confirming that imaginary attention plays a more dominant role in long context modeling.

## 5.3 COMBINATION WITH OTHER LONG-CONTEXT TECHNIQUES

RoPE++ can not only be combined with NTK for context extension during long-context training, but can also be combined with other long-context techniques such as Linear PI (Chen et al., 2023) and YaRN (Peng et al., 2024). Across 376M and 776M model sizes, we conduct extensive experiments of long-context further pre-training in 32k context length, with the interpolation coefficient $s = 8$ for Linear PI and $s = 32$ for YaRN, the default values in the original paper. The results are shown in Table 3. We report the perplexity on WikiText and the average score of tasks we have presented in Table 1 as the summary of short-context performance, with the full results in Table 10. Results show that RoPE++ consistently achieves the highest scores on RULER, BABILong, and short-context average score, confirming its advantage and generalization. More analysis on larger model scale and training convergence is detailed in Appendix C. More discussion on the extrapolation performance and limitation of RoPE++ can be found in Appendix D.

## 6 CONCLUSION

We introduce RoPE++, which employs both real and imaginary attentions. Mathematical analysis first reveals the imaginary attention's potential for modeling long-context dependencies. Building upon this, we re-incorporate the originally discarded imaginary attention as a new group of heads while preserving the unified absolute–relative position embedding format. Particularly, we introduce RoPE++$_{EH}$, with equal head as well as halved cache, and RoPE++$_{EC}$ with equal cache and doubled heads. Pre-training and evaluation at 376M and 776M model sizes show that both RoPE++$_{EH}$ and RoPE++$_{EC}$ outperform vanilla RoPE and other position embeddings on average across short-context tasks and acquire even larger gains in long-context scenarios. Further analysis confirms that imaginary attentions are more dominant in long-context modeling compared with original real attention, validating their effectiveness in enhancing long-context LLMs.

ACKNOWLEDGEMENT

This work was supported by the National Natural Science Foundation of China (No. U24B20181) and Shanghai Pilot Program for Basic Research - Fudan University 21TQ1400100 (22TQ018). We greatly appreciate all reviewers for their constructive reviews, and thanks to Jiasheng Ye for the discussion on scaling verification of model architecture.

ETHICAL STATEMENT

This research follows established ethical standards and practice principles. To our knowledge, our study processes no sensitive personal data, involves no human subjects, and targets no ethically risky applications. All experiments and analyses comply with recognized guidelines, ensuring integrity, transparency, and reliability.

REPRODUCIBILITY STATEMENT

To ensure the reproducibility of and to support the open-source community, we have publicly released RoPE++, its trained checkpoints, and the complete training and evaluation code. We expect these as a reference for future work on long-context LLMs, facilitating progress in this field.

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

## A USE OF LARGE LANGUAGE MODELS

We use Large Language Models solely for language-centric assistance, including checking grammar, style, and clarity. No aspect of research, including ideation, experimental design, or scientific contribution, is influenced or generated by the output of LLMs.

## B PRELIMINARY: KEY PROPERTIES OF ROPE

Rotary Position Embedding (RoPE) encodes absolute positions by splitting the feature dimensions of query and key vectors $\boldsymbol{q}_t, \boldsymbol{k}_s$ into 2-D pairs and rotating each pair (Su et al., 2024). The rotation angle is the product of the token index $t$ or $s$, and $\theta_n$. Owing to the properties of rotation matrices, the independently applied absolute position embedding on $\boldsymbol{q}_t, \boldsymbol{k}_s$ fuse into a relative position embedding, namely $\cos\theta_n(t - s), \sin\theta_n(t - s)$, of the attention matrix, as shown in Equation 6.

$$
\begin{aligned}
\boldsymbol{A}_{t,s} &= \underbrace{\sum_{n=0}^{d/2-1} \begin{bmatrix} q_t^{(2n)} \\ q_t^{(2n+1)} \end{bmatrix}^\top \begin{bmatrix} \cos\theta_n(t-s) & \sin\theta_n(t-s) \\ -\sin\theta_n(t-s) & \cos\theta_n(t-s) \end{bmatrix} \begin{bmatrix} k_s^{(2n)} \\ k_s^{(2n+1)} \end{bmatrix}}_{\text{Relative PE}} \\
&= \underbrace{\sum_{n=0}^{d/2-1} \left( \begin{bmatrix} \cos\theta_n t & -\sin\theta_n t \\ \sin\theta_n t & \cos\theta_n t \end{bmatrix} \begin{bmatrix} q_t^{(2n)} \\ q_t^{(2n+1)} \end{bmatrix} \right)^\top \left( \begin{bmatrix} \cos\theta_n s & -\sin\theta_n s \\ \sin\theta_n s & \cos\theta_n s \end{bmatrix} \begin{bmatrix} k_s^{(2n)} \\ k_s^{(2n+1)} \end{bmatrix} \right)}_{\text{Absolute PE}}
\end{aligned}
\tag{6}
$$

By default, the rotary angles $\theta_n = 10000^{-2n/d}, n = 0, \cdots, d/2 - 1$.

Equation 6 presents RoPE in vector form. Since any 2-D vector corresponds to a complex number, the rotation of such a vector is equivalent to complex multiplication.

$$
\tilde{q}_t^{(n)} = q_t^{(2n)} + i \cdot q_t^{(2n+1)}, \quad \tilde{k}_s^{(n)} = k_s^{(2n)} + i \cdot k_s^{(2n+1)}
$$

Building on this equivalence, RoPE can be expressed in complex form as shown in Equation 1.

Besides unifying relative and absolute position embeddings, RoPE exhibits semantic aggregation and long-context decay (Su et al., 2024). On one hand, when $\boldsymbol{q}, \boldsymbol{k}$ vectors are semantically close, their attention score remains large on average, regardless of relative distance $\Delta t$. This property is detailed in Su (2024b). If we have a vector $\boldsymbol{k}$ that is independent and identically distributed with respect to $\boldsymbol{q}$, with average $\mu$ and variance $\sigma^2$ for every feature dimension, and a vector that is only slightly perturbed with respect to $\boldsymbol{q} + \boldsymbol{\varepsilon}$, the expected attention score difference can be calculated as follows and proved to be positive.

$$
\begin{aligned}
&\mathbb{E}_{\boldsymbol{q},\boldsymbol{k},\boldsymbol{\varepsilon}} \left[ \boldsymbol{q}^\top \boldsymbol{\mathcal{R}}_{-\Delta t}(\boldsymbol{q} + \boldsymbol{\varepsilon}) - \boldsymbol{q}^\top \boldsymbol{\mathcal{R}}_{-\Delta t} \boldsymbol{k} \right] \\
=& \mathbb{E}_{\boldsymbol{q}} \left[ \boldsymbol{q}^\top \boldsymbol{\mathcal{R}}_{-\Delta t} \boldsymbol{q} \right] - \mathbb{E}_{\boldsymbol{q},\boldsymbol{k}} \left[ \boldsymbol{q}^\top \boldsymbol{\mathcal{R}}_{-\Delta t} \boldsymbol{k} \right] \\
=& \mathbb{E}_{\boldsymbol{q}} \left[ \boldsymbol{q}^\top \boldsymbol{\mathcal{R}}_{-\Delta t} \boldsymbol{q} \right] - \mathbb{E}_{\boldsymbol{q}}[\boldsymbol{q}]^\top \boldsymbol{\mathcal{R}}_{-\Delta t} \mathbb{E}_{\boldsymbol{k}}[\boldsymbol{k}] \\
=& \mathbb{E}_{\boldsymbol{q}} \left[ \boldsymbol{q}^\top \boldsymbol{\mathcal{R}}_{-\Delta t} \boldsymbol{q} \right] - \mu^2 \mathbf{1}^\top \boldsymbol{\mathcal{R}}_{-\Delta t} \mathbf{1} \\
=& \mathbb{E}_{\boldsymbol{q}} \left[ \sum_{n=0}^{d/2-1} \left( q^{(2n)^2} + q^{(2n+1)^2} \right) \cos(-\theta_n \Delta t) \right] - \sum_{n=0}^{d/2-1} 2\mu^2 \cos(-\theta_n \Delta t) \\
=& \sum_{n=0}^{d/2-1} 2\left( \mu^2 + \sigma^2 \right) \cos\theta_n \Delta t - \sum_{n=0}^{d/2-1} 2\mu^2 \cos\theta_n \Delta t \\
=& \sum_{n=0}^{d/2-1} 2\sigma^2 \cos\left( 10000^{-\frac{2n}{d}} \Delta t \right) > 0
\end{aligned}
$$

On the other hand, as $\Delta t$ increases, the attention between any $\boldsymbol{q}, \boldsymbol{k}$ decreases on average. We can similarly derive this by showing that the expectation of the attention score, as shown below, is almost

monotonically decaying with the increase of $\Delta t$.

$$\mathbb{E}_{\boldsymbol{q},\boldsymbol{k}}\left[\boldsymbol{q}^\top \boldsymbol{\mathcal{R}}_{-\Delta t}\boldsymbol{k}\right] = \mathbb{E}_{\boldsymbol{q}}\left[\sum_{n=0}^{d/2-1}\left(q^{(2n)}k^{(2n)}+q^{(2n+1)}k^{(2n+1)}\right)\cos\left(\theta_n\Delta t\right)\right]$$

$$= \sum_{n=0}^{d/2-1} 2(\mu^2+\sigma^2)\cos\left(10000^{-\frac{2n}{d}}\Delta t\right)$$

Both properties arise from averaging $\cos(\theta\Delta t)$ over frequency $\theta$ sampled based on $\theta_n = 10000^{-2n/d}, n = 0, \cdots, d/2-1$. It is a discrete approximation $c_{\mathrm{Re}}(\Delta t)$ to a cosine integral function $\tilde{c}_{\mathrm{Re}}(\Delta t)$, as shown in Equation 7. We refer to this as the characteristic curve of RoPE, as shown in Figure 1. It is positive and decaying, conferring these two mathematical properties of RoPE.

$$c_{\mathrm{Re}}(\Delta t) = \frac{2}{d}\sum_{n=0}^{d/2-1}\cos\left(10^{-\frac{8n}{d}}\Delta t\right), \quad \tilde{c}_{\mathrm{Re}}(\Delta t) = \int_{10^{-4}}^{1}\frac{\cos\theta t}{\theta\ln 10^4}\mathrm{d}\theta = \mathrm{Ci}(\Delta t) - \mathrm{Ci}\left(\frac{\Delta t}{10^4}\right) \quad (7)$$

For the imaginary part lost in RoPE's complex representation, we derive a sine integral function, and that is the characteristic curve for the imaginary attention in RoPE++, as shown in Equation 5.

$$\mathbb{E}_{\boldsymbol{q},\boldsymbol{k},\boldsymbol{\varepsilon}}\left[\boldsymbol{q}^\top\boldsymbol{\mathcal{R}}_{-\frac{\pi}{2}-\Delta t}(\boldsymbol{q}+\boldsymbol{\varepsilon}) - \boldsymbol{q}^\top\boldsymbol{\mathcal{R}}_{-\frac{\pi}{2}-\Delta t}\boldsymbol{k}\right] = \sum_{n=0}^{d/2-1} 2\sigma^2\sin\left(10000^{-\frac{2n}{d}}\Delta t\right) > 0$$

$$\mathbb{E}_{\boldsymbol{q},\boldsymbol{k}}\left[\boldsymbol{q}^\top\boldsymbol{\mathcal{R}}_{-\frac{\pi}{2}-\Delta t}\boldsymbol{k}\right] = \sum_{n=0}^{d/2-1} 2(\mu^2+\sigma^2)\sin\left(10000^{-\frac{2n}{d}}\Delta t\right)$$

$$c_{\mathrm{Im}}(\Delta t) = \frac{2}{d}\sum_{n=0}^{d/2-1}\sin\left(10^{-\frac{8n}{d}}\Delta t\right), \quad \tilde{c}_{\mathrm{Im}} = \int_{10^{-4}}^{1}\frac{\sin\theta t}{\theta\ln 10^4}\mathrm{d}\theta = \mathrm{Si}(\Delta t) - \mathrm{Si}\left(\frac{\Delta t}{10^4}\right)$$

Finally, we should also clarify that our analysis is expectation-based, i.e., the average fluctuation of the real or imaginary attention, which is different from the case-study-level discussion in p-RoPE (Barbero et al., 2024). The analysis of p-RoPE on long-context decay is orthogonal to our contribution. We can likewise drop the rotary angles for the low-frequency dimensions in both the real and imaginary attention of RoPE++.

## C   MORE EXPERIMENT

The configuration of our 376M, 776M and 1.5B models can be summarized in the following table. Our models use the same tokenizer as the Llama 3 Series (Meta, 2024b;a; Dubey et al., 2024).

|  | 376M | 776M | 1.5B |
| --- | --- | --- | --- |
| Hidden Size | 1024 | 1536 | 2048 |
| Intermediate Size | 3584 | 5376 | 7168 |
| Num Layer | 8 | 12 | 16 |
| Num Attn Head | 8 | 12 | 16 |
| Num KV Head | 4 | 6 | 4 |
| Vocab Size | 128256 | 128256 | 128256 |

Table 4: The hyper-parameter of different model sizes.

### C.1   VALIDATION ON LARGER SCALE

Scaling validation is essential for architectural research, though concurrent or earlier work still only performs pre-training validation on models smaller than 1B and data volumes smaller than

| | Wiki ppl ↓ | LMB ppl ↓ | TQA acc ↑ | PIQA acc ↑ | Hella acc ↑ | Wino acc ↑ | ARC-e acc ↑ | GPQA acc ↑ | SIQA acc ↑ | OBQA acc ↑ | SG acc ↑ | Avg. |
|---|---|---|---|---|---|---|---|---|---|---|---|---|
| ***1.5B Short*** | | | | | | | | | | | | |
| RoPE | 27.9 | **15.3** | 36.4 | 70.8 | 46.2 | 53.2 | 44.8 | 25.8 | 39.4 | 24.6 | 44.8 | 42.9 |
| RoPE++$_{EH}$ | 28.0 | 16.0 | 37.1 | 71.2 | 45.6 | 53.6 | 45.9 | 25.8 | 40.8 | 27.8 | 44.3 | **43.6** |
| RoPE++$_{EC}$ | **27.4** | 15.5 | 35.1 | 69.5 | 46.3 | 53.3 | 43.6 | 28.3 | 40.5 | 23.6 | 45.6 | 42.9 |
| ***1.5B Long*** | | | | | | | | | | | | |
| RoPE | 24.8 | 12.5 | 36.3 | 71.1 | 49.3 | 56.7 | 48.9 | 27.3 | 40.8 | 24.2 | 46.1 | **44.5** |
| RoPE++$_{EH}$ | 25.1 | 13.0 | 36.1 | 71.3 | 48.0 | 55.9 | 48.0 | 23.2 | 41.2 | 29.2 | 44.1 | 44.1 |
| RoPE++$_{EC}$ | **24.4** | **12.4** | 36.6 | 71.3 | 49.6 | 55.3 | 44.8 | 20.7 | 41.1 | 26.2 | 45.1 | 43.4 |

Table 5: Results on short-context tasks for 1.5B models.

| | RULER | | | | | | BABILong | | | | | | |
|---|---|---|---|---|---|---|---|---|---|---|---|---|---|
| | 4k | 8k | 16k | 32k | 64k | Avg. | 2k | 4k | 8k | 16k | 32k | 64k | Avg. |
| RoPE | 50.5 | 39.9 | 34.6 | **31.9** | 18.3 | 35.1 | 25.9 | 29.7 | **31.8** | **30.5** | **18.5** | 13.5 | 29.5 |
| RoPE++$_{EH}$ | 42.5 | 38.1 | 33.1 | 28.7 | 12.7 | 31.0 | **40.0** | **35.9** | 30.5 | 25.1 | 14.7 | **15.8** | **32.9** |
| RoPE++$_{EC}$ | **53.1** | **45.7** | **39.0** | 30.6 | **18.9** | **37.5** | 25.1 | 28.3 | 21.4 | 16.6 | 12.6 | 12.3 | 22.9 |

Table 6: Results on long-context tasks for 1.5B models further trained with 5B tokens in 32k.

| | 5B | 10B | 15B | 20B | 30B | 40B | 50B |
|---|---|---|---|---|---|---|---|
| ***Training Loss*** | | | | | | | |
| RoPE | 3.4698 | 3.3576 | 3.2249 | 3.3459 | 3.2945 | 3.2946 | 3.1665 |
| RoPE++$_{EH}$ | 3.4904 | 3.3821 | 3.2511 | 3.3708 | 3.3202 | 3.3194 | 3.1931 |
| RoPE++$_{EC}$ | 3.4567 | 3.3473 | 3.2203 | 3.3373 | 3.2848 | 3.2904 | 3.1612 |
| ***Validation Loss*** | | | | | | | |
| RoPE | 3.5509 | 3.4358 | 3.3933 | 3.3629 | 3.3299 | 3.3177 | 3.1881 |
| RoPE++$_{EH}$ | 3.5772 | 3.4618 | 3.4217 | 3.3907 | 3.3567 | 3.3430 | 3.2141 |
| RoPE++$_{EC}$ | 3.5362 | 3.4254 | 3.3870 | 3.3569 | 3.3251 | 3.3140 | 3.2683 |
| ***Average Score*** | | | | | | | |
| RoPE | **39.2** | 39.6 | **40.4** | **39.9** | 39.5 | 39.8 | 40.1 |
| RoPE++$_{EH}$ | 38.2 | **39.6** | 40.3 | 38.9 | 39.5 | 39.8 | 40.3 |
| RoPE++$_{EC}$ | 38.8 | 38.8 | 39.3 | 39.6 | **40.3** | **40.2** | **41.0** |

Table 7: Comparison of training loss, validation loss, and the average score of short-context tasks between RoPE and RoPE++ on the 376M model size under different training tokens.

50B tokens (Dai et al., 2025; Hua et al., 2024). Unfortunately, our available resources limit us to scales below 7B. After an extended effort, we have completed a 1.5B model trained on 50B tokens in 4k context length, followed by 5B tokens in 32k context length, using the same training hyperparameters used for 776M and 376M. The results in Table 5 and Table 6 demonstrate that RoPE++ still outperforms RoPE. To sum up, within the limits of our available computing resources, we have successfully validated the effectiveness of RoPE++ across all three model scales.

Concerning the data scales, based on the experiments on 776M and 376M, 50B tokens are already sufficient for convergence at this scale, as evidenced by plateaued training loss, validation loss, and average short-context scores in Table 7. We verify this judgment by evaluating 776M and 1.5B checkpoints pre-trained on more tokens while the learning rate remains constant. Beyond 50B tokens, the model shows no significant further gain. These results also demonstrate that RoPE++ exhibits loss curves that almost overlap with those of RoPE, showing no training stability issues. Although RoPE may show an advantage in the early checkpoints, RoPE++ continues to improve and ultimately surpasses RoPE in average scores. Note that these 776M results at 50B tokens are obtained without learning-rate annealing, so they differ slightly from the scores reported above.

|  | 5B | 10B | 15B | 20B | 30B | 40B | 50B | 60B | 70B |
|---|---|---|---|---|---|---|---|---|---|
| *Training Loss* | | | | | | | | | |
| RoPE | 3.2826 | 3.1621 | 3.0279 | 3.1418 | 3.0889 | 3.0913 | 3.0502 | 3.0208 | 3.0307 |
| + RoPE++$_{EH}$ | 3.3104 | 3.1955 | 3.0622 | 3.1701 | 3.1236 | 3.1237 | 3.0798 | 3.0496 | 3.0596 |
| + RoPE++$_{EC}$ | 3.2797 | 3.1620 | 3.0294 | 3.1377 | 3.0890 | 3.0889 | 3.0489 | 3.0200 | 3.0294 |
| *Validation Loss* | | | | | | | | | |
| RoPE | 3.3480 | 3.2279 | 3.1842 | 3.1552 | 3.1214 | 3.1083 | 3.0912 | 3.0817 | 3.0751 |
| + RoPE++$_{EH}$ | 3.3865 | 3.2648 | 3.2184 | 3.1877 | 3.1545 | 3.1401 | 3.1232 | 3.1168 | 3.1050 |
| + RoPE++$_{EC}$ | 3.3541 | 3.2305 | 3.1872 | 3.1550 | 3.1205 | 3.1061 | 3.0903 | 3.0816 | 3.0733 |
| *Average Score* | | | | | | | | | |
| RoPE | 39.3 | 40.8 | **41.3** | **41.5** | 41.4 | 41.5 | **41.7** | **42.3** | 41.8 |
| + RoPE++$_{EH}$ | **40.0** | **40.9** | 41.2 | 40.6 | 41.7 | 42.2 | 41.3 | 41.9 | **42.1** |
| + RoPE++$_{EC}$ | 38.7 | 40.4 | 41.1 | 41.0 | **41.8** | 41.7 | **41.7** | 41.8 | 41.9 |

Table 8: Comparison of training loss, validation loss, and the average score of short-context tasks between RoPE and RoPE++ on the 776M model size under different training tokens.

|  | 5B | 10B | 15B | 20B | 30B | 40B | 50B | 60B | 70B | 80B | 90B | 100B |
|---|---|---|---|---|---|---|---|---|---|---|---|---|
| *Training Loss* | | | | | | | | | | | | |
| RoPE | 3.1798 | 3.0462 | 2.9492 | 2.9048 | 2.9835 | 2.8967 | 2.9447 | 2.8384 | 2.9855 | 2.8224 | 2.9697 | 2.9775 |
| RoPE++$_{EH}$ | 3.2051 | 3.0718 | 2.9724 | 2.9333 | 3.0104 | 2.9244 | 2.9688 | 2.8619 | 3.0110 | 2.8491 | 2.9980 | 3.0018 |
| RoPE++$_{EC}$ | 3.1708 | 3.0384 | 2.9418 | 2.9025 | 2.9824 | 2.8916 | 2.9404 | 2.8307 | 2.9775 | 2.8177 | 2.9664 | 2.9736 |
| *Validation Loss* | | | | | | | | | | | | |
| RoPE | 3.2498 | 3.1297 | 3.0754 | 3.0492 | 3.0112 | 2.9911 | 2.8477 | 2.9711 | 2.9641 | 2.9526 | 2.9503 | 2.9464 |
| RoPE++$_{EH}$ | 3.2768 | 3.1569 | 3.1021 | 3.0785 | 3.0406 | 3.0167 | 2.8744 | 2.9949 | 2.9904 | 2.9799 | 2.9780 | 2.9681 |
| RoPE++$_{EC}$ | 3.2420 | 3.1236 | 3.0709 | 3.0466 | 3.0086 | 2.9857 | 2.8456 | 2.9669 | 2.9577 | 2.9515 | 2.9488 | 2.9433 |
| *Average Score* | | | | | | | | | | | | |
| RoPE | **40.9** | **41.5** | **42.2** | **42.7** | **43.0** | **43.2** | 42.9 | 42.7 | **43.2** | **43.8** | 43.4 | 43.0 |
| RoPE++$_{EH}$ | 39.6 | 40.8 | 41.8 | 41.9 | 42.1 | 42.5 | **43.6** | **43.3** | 42.4 | 43.4 | 42.8 | 42.7 |
| RoPE++$_{EC}$ | 39.6 | 41.2 | 41.4 | 42.0 | 42.7 | 42.4 | 42.9 | 42.7 | 42.7 | 43.0 | **43.8** | **43.1** |

Table 9: Comparison of training loss, validation loss, and the average score of short-context tasks between RoPE and RoPE++ on the 1.5B model size under different training tokens.

## C.2 MORE DETAILED RESULTS

The detailed results on short-context tasks for 776M and 376M models further trained with 5B tokens in 32k context length with YaRN and Linear PI are shown in Table 10. We also add the comparison of the training throughput (TGS, tokens per GPU per second) at 4k and 32k training context lengths, as well as the model storage (GB), as shown in Table 11. Notably, though RoPE$_{EC}$ keeps the cache size fixed, it still increases computation and the size of $W_o$. Nevertheless, long-context inference is primarily IO-bounded rather than computation-bounded and dominated by KV cache memory cost. Therefore, the absence of additional cache overhead remains acceptable. We will continue to develop more elegant computation optimizations tailored to RoPE++ to reduce this additional cost.

## D MORE DISCUSSION

### D.1 POTENTIAL REDUNDANCY AND CONFLICT

Regarding possible redundancy and conflict among attention heads, we first clarify that these issues already exist in vanilla RoPE-based LLMs, motivating works that directly compress KV cache (such as MLA (Liu et al., 2024a), GQA (Ainslie et al., 2023)) or distinguish head types (such as DuoAttention (Xiao et al., 2024b), MInference (Jiang et al., 2024)). Therefore, this redundancy and conflict likewise remain present in RoPE++. Concerning redundancy, the imaginary and real attentions exhibit distinct biases and, as shown in Figure 5, show different functional patterns. Because RoPE++$_{EH}$ outperforms vanilla RoPE while using half the cache, redundancy between the

| | Wiki | LMB | TQA | PIQA | Hella | Wino | ARC-e | GPQA | SIQA | OBQA | SG | Avg. |
|---|---|---|---|---|---|---|---|---|---|---|---|---|
| | ppl ↓ | ppl ↓ | acc ↑ | acc ↑ | acc ↑ | acc ↑ | acc ↑ | acc ↑ | acc ↑ | acc ↑ | acc ↑ | |
| *376M Long PI* | | | | | | | | | | | | |
| RoPE | **33.4** | 20.1 | 34.9 | 69.4 | 43.1 | 52.5 | 42.9 | 29.3 | 40.0 | 21.6 | 44.3 | 42.0 |
| RoPE++$_{EH}$ | 34.7 | 20.7 | 35.5 | 68.4 | 41.1 | 53.7 | 44.6 | 22.7 | 40.0 | 26.0 | 42.9 | 41.7 |
| RoPE++$_{EC}$ | 33.7 | **19.4** | 36.6 | 70.1 | 43.1 | 51.9 | 44.8 | 27.3 | 40.7 | 27.0 | 44.1 | **42.8** |
| *376M Long YaRN* | | | | | | | | | | | | |
| RoPE | **32.8** | 19.3 | 36.1 | 69.9 | 43.5 | 52.6 | 44.1 | 25.3 | 41.2 | 24.0 | 43.4 | 42.2 |
| RoPE++$_{EH}$ | 33.9 | 20.1 | 35.7 | 68.8 | 41.8 | 55.0 | 45.2 | 25.8 | 40.4 | 24.6 | 42.2 | 42.2 |
| RoPE++$_{EC}$ | 32.9 | **18.8** | 36.7 | 69.5 | 43.6 | 52.6 | 44.1 | 29.3 | 41.0 | 30.2 | 44.0 | **43.4** |
| *776M Long PI* | | | | | | | | | | | | |
| RoPE | **27.8** | **14.8** | 35.8 | 65.4 | 33.8 | 53.1 | 38.8 | 25.3 | 39.6 | 27.2 | 44.7 | 40.4 |
| RoPE++$_{EH}$ | 28.8 | 15.6 | 37.0 | 66.0 | 33.1 | 52.6 | 40.2 | 25.8 | 38.1 | 26.4 | 44.0 | 40.4 |
| RoPE++$_{EC}$ | **27.8** | 14.9 | 37.0 | 65.4 | 33.9 | 51.7 | 39.5 | 23.2 | 39.4 | 28.4 | 45.8 | **40.5** |
| *776M Long YaRN* | | | | | | | | | | | | |
| RoPE | 27.3 | 14.5 | 36.7 | 65.6 | 33.9 | 51.1 | 37.7 | 29.3 | 39.7 | 27.6 | 46.3 | 40.9 |
| RoPE++$_{EH}$ | 28.3 | 15.1 | 38.0 | 66.1 | 33.6 | 51.6 | 40.9 | 26.3 | 39.3 | 26.0 | 43.9 | 40.6 |
| RoPE++$_{EC}$ | 27.3 | 14.5 | 37.1 | 65.8 | 34.8 | 53.4 | 41.8 | 27.8 | 40.1 | 27.2 | 45.1 | **41.5** |

Table 10: Results on short-context tasks for 776M and 376M models further trained with 5B tokens in 32k context length with YaRN and Linear PI.

| | 376M | | | 776M | | | 1.5B | | |
|---|---|---|---|---|---|---|---|---|---|
| | TGS-4k | TGS-32k | Storage | TGS-4k | TGS-32k | Storage | TGS-4k | TGS-32k | Storage |
| RoPE | 80248.2 | 53317.4 | 0.8 | 49617.2 | 29019.6 | 1.6 | 32497.2 | 17040.8 | 2.7 |
| RoPE++$_{EH}$ | 80248.2 | 53498.8 | 0.7 | 50574.4 | 29399.3 | 1.5 | 33752.4 | 17672.6 | 2.6 |
| RoPE++$_{EC}$ | 70457.5 | 37271.7 | 0.8 | 44431.2 | 22631.1 | 1.6 | 26479.2 | 10922.7 | 2.9 |

Table 11: The comparison of the throughput (TGS, tokens per GPU per second) at 4k and 32k training context lengths, as well as the model storage (GB).

two components is lower than that among standard heads. Regarding conflict, although each query vector in RoPE++ participates in both real and imaginary attention, the superior results of RoPE++$_{EC}$ over vanilla RoPE under identical cache size shown in Table 1 and Table 2 indicate that the benefits of this potential conflict outweigh its possible drawbacks.

### D.2 PERPLEXITY CURVE OF ROPE++

Length extrapolation is a central issue for long-context LLMs. We have already shown that RoPE++ outperforms RoPE on long-context downstream tasks in training-based length extrapolation. However, RoPE++ cannot directly extrapolate like FoPE (Hua et al., 2024) or PaTH (Yang et al., 2025b). Once the inference exceeds the maximum supported context length, perplexity begins to rise. Interestingly, as discussed in Section 3.4, every even-index dimension in query vectors and odd-index dimension in key vectors are trained with full value range of position embeddings, and every dimension has seen both positive and negative positions during training. Consequently, the perplexity curve of RoPE++ climbs more gradually (Liu et al., 2024d).

This is verified in Figure 6, where we compare the perplexity of short-context-trained RoPE and RoPE++ on 376M and 776M model sizes. With or without fixed-NTK interpolation based on scaling factor $\lambda = 4$, both curves rise at the same context length with RoPE, indicating an identical stable context upper bound. Beyond that point, however, RoPE++'s perplexity increases more slowly, confirming the earlier prediction about its extrapolation behavior.

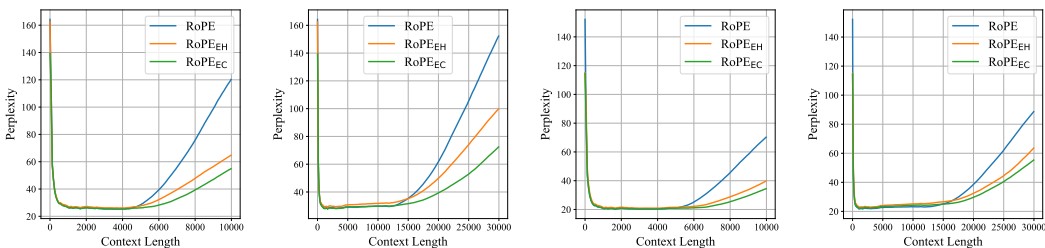

(a) Perplexity curve of pre-trained 376M.

(b) Perplexity curve of NTK-scaled 376M.

(c) Perplexity curve of pre-trained 776M.

(d) Perplexity curve of NTK-scaled 776M.

Figure 6: Perplexity comparison between RoPE and RoPE++ in 376M and 776M models.

## LIMITATION

As noted above, RoPE++ markedly boosts performance on both short- and long-context tasks, yet it needs training from scratch and fails to deliver plug-and-play length extrapolation, falling behind such extrapolation designs as FoPE and PaTH. Nevertheless, as a method that reintroduces imaginary attention, raising performance under fixed memory or improving efficiency while preserving accuracy, RoPE++ can be integrated with part of those designs. Additionally, thanks to the oddity of the sine function, the imaginary component also shows promise for bidirectional-attention-based diffusion language models (Nie et al., 2025; Ye et al., 2025) as well as its extrapolation (Liu et al., 2025b), and we will provide experiments on these aspects in follow-up.

