# OpenReview forum: "Beyond Real: Imaginary Extension of Rotary Position Embeddings for Long-Context LLMs"
_ICLR.cc/2026/Conference — ICLR 2026 Poster_

### Official Review · Reviewer_mcxi · 2025-10-24

**Soundness:** 3
**Presentation:** 3
**Contribution:** 3
**Rating:** 8
**Confidence:** 3

**Summary:**

This paper identifies that standard Rotary Position Embeddings (RoPE) discard the imaginary component of complex-valued dot products during attention score calculation(this component potentially containing valuable phase information relevant to long-range dependency modeling). To address this limitation, the authors propose **RoPE++**, an extension that re-incorporates the imaginary component to enable a full complex-valued representation of attention scores, forming dual-component (real + imaginary) attention.

**Strengths:**

1. The proposed **RoPE++$_{EH}$** configuration achieves remarkable efficiency: it maintains the same number of attention heads as vanilla RoPE while halving KV cache and QKV parameters, which is highly valuable for long-context LLMs where memory constraints are critical.
2. Empirically, RoPE++ demonstrates consistent advantages over vanilla RoPE across both short-context tasks (e.g., WikiText perplexity, Open LLM Leaderboard benchmarks) and long-context tasks (e.g., RULER, BABILong up to 64k context), validating its generality and effectiveness.

**Weaknesses:**

1. RoPE++ requires full pre-training (or continuous long-context pre-training) to realize its performance gains and cannot be applied as a "plug-and-play" module—this limits its flexibility for scenarios where re-training is computationally prohibitive or unavailable.
2. The **RoPE++$_{EC}$** configuration, while delivering superior long-context performance, doubles the number of attention heads (relative to vanilla RoPE) under fixed KV cache size, introducing additional computational overhead that may offset its performance benefits in resource-constrained settings.

**Questions:**

1. In Section 3.2, the authors argue that the imaginary component of RoPE++ attends to more distant positions, and Figure 1 supports this by showing that the imaginary attention’s magnitude gradually becomes more prominent for Δt > 25 (facilitating long-range dependency modeling). However, the curve exhibits unexpected behavior near Δt = 0 (a sharp transition) and for Δt < 25 (magnitude first decreases then increases). Could the authors elaborate on whether this short-range fluctuation affects RoPE++’s ability to preserve semantic aggregation ? For example, does it disrupt local semantic coherence when modeling short-range token relationships?
2. For RoPE++$_{EC}$: A single query vector (q) undergoes two positional encodings (one for real attention, one for imaginary attention) before computing dot products with keys (k). Do the authors observe any information representation conflicts between the two sets of dot products? If so, how does the model resolve such conflicts to avoid degrading attention quality? If not, what mechanisms (e.g., parameter sharing, attention head specialization) ensure complementary rather than redundant information from the two encodings?
3. Existing RoPE extension methods (e.g., NTK-PI, YaRN, LongRoPE) modify RoPE’s behavior via dimension-wise adjustments (e.g., scaling rotary bases, interpolating index ranges, partitioning feature dimensions) to enhance length extrapolation. Could the authors clarify whether these dimension-level modifications are compatible with RoPE++’s imaginary component extension? For instance, do such methods introduce conflicts in how positional information is encoded across real/imaginary components, or can they be combined to further improve long-context performance?

---

> ### Author Response · Authors · 2025-11-25
> **Reply (3/3)**
>
> First, we sincerely appreciate the reviewer for pointing out our remarkable efficiency, recognizing consistent advantages over traditional RoPE design, and raising such thoughtful reviews on position embeddings.
>
> **Regarding W1 on no plug-and-play**
>
> Thank you for raising this point. Our method indeed requires pre-training from scratch and cannot be integrated directly. Nevertheless, this does not diminish its key contribution. We identify that RoPE discards the imaginary component of its complex calculation and show that re-incorporating this component improves downstream performance even when halving cache and attention-head parameters. From this perspective, plug-and-play integration is not a prerequisite property for RoPE++ and its related work. The concurrent HoPE [1] and DoPE [2], the earlier Partial RoPE adopted by Pythia [3], and the even earlier ALiBi [4] all need pre-training from scratch. In fact, RoPE itself is also proposed as an enhancement of sinusoidal position embedding and is not a plug-and-play solution.
>
> **Regarding W2 on additional computational overhead**
>
> Thank you for the question. RoPE++ offers two variants. RoPE$\_\text{EH}$ maintains the same head number and halves the KV cache and QKV parameters, which is highly valuable as the reviewer has recognized for long-context LLMs where memory is critical. RoPE$\_\text{EC}$ keeps the cache size fixed, slightly increasing computation and O parameters without adding extra cache. Nevertheless, long-context inference is primarily IO-bounded rather than computation-bounded and dominated by KV cache memory cost. Therefore, the absence of additional cache overhead remains acceptable. We will continue to develop more elegant computation optimizations tailored to RoPE++ to reduce this additional computational cost.
>
> **Regarding Q1 on the property of the imaginary component**
>
> Thank you for your insightful question. In Section 5.2, we analyze the attention pattern of RoPE++ and find that the imaginary heads show no anomaly caused by the short-range fluctuations of the characteristic curve. Like standard heads, they focus primarily on the initial and local tokens, and practical evaluation confirms fluent generation. Owing to the bias introduced by the characteristic curve, imaginary attention pays even greater attention to the beginning of the context (Figure 5a–d, f–i), a pattern that earlier studies [5-6] have visualized and linked to a stronger capacity for capturing long-distance semantic dependencies, a claim we corroborate in Figure 5e,j. To further illuminate the implications of the imaginary heads, the revised version will add case studies that visualize how real and imaginary attention scores differ between RoPE++ and RoPE when generating distinct answers. We thank the reviewer again for this constructive question.
>
> **Regarding Q2 on information representation redundancy and conflict**
>
> We thank you for the insightful question and are glad that the principle and implementation of RoPE++ are clearly understood. Regarding possible redundancy and conflict among attention heads, we first clarify that these issues already exist in vanilla RoPE-based LLMs, motivating works that directly compress KV cache (such as MLA, GQA [7-8]) or distinguish head types (such as DuoAttention, MInference [6,9]). Therefore, this redundancy and conflict likewise remain present in RoPE++. Concerning redundancy, the imaginary and real attentions exhibit distinct biases and, as shown in Figure 5, show different functional patterns. Because RoPE$\_\text{EH}$ outperforms vanilla RoPE while using half the cache, redundancy between the two components is lower than that among standard heads. Concerning conflict, although each query vector in RoPE++ participates in both real and imaginary attention, the superior results of RoPE$\_\text{EC}$ over vanilla RoPE under identical cache size indicate that the benefits of this potential conflict outweigh its possible drawbacks. We appreciate the reviewer’s questions and will incorporate this analysis into the revised version to clarify our mechanism and advantages.

---

> ### Author Response · Authors · 2025-11-25
> **Reply (2/3)**
>
> **Regarding Q3 on combining with other long-context techniques**
>
> Thank you for this constructive review. RoPE++ can not only be combined with NTK for context extension during long-context training, but can also be combined with Linear PI[10] and YaRN[11]. Across 376M and 776M model sizes, we conduct extensive experiments, with the interpolation coefficient $s = 8$ for Linear PI and $s = 32$ for YaRN. RoPE++ consistently achieves the highest composite scores on RULER, BABILong, and the average score of short-context tasks, confirming its advantage and generalization. These results will be added to the revised paper.
>
> |                      | RU    | LE    | R     |       |       | BA    | BI    | Lo    | ng    |       |       |
> |:---------------------|:-----:|:-----:|:-----:|:-----:|:-----:|:-----:|:-----:|:-----:|:-----:|:-----:|:-----:|
> |                      | 4k    | 8k    | 16k   | 32k   | Avg   | 2k    | 4k    | 8k    | 16k   | 32k   | Avg   |
> |***776M Long***       |       |       |       |       |       |       |       |       |       |       |       |
> | RoPE + NTK           | 37.4  | 35.1  | 33.0  | 21.2  | 31.7  | 33.5  | 30.7  | 23.6  | 22.0  | 15.1  | 25.0  |
> | + RoPE++$_\text{EH}$ | 38.7  | 35.4  | 33.8  | 24.6  | 33.1  | 31.9  | 26.5  | 18.6  | 16.2  | 11.0  | 20.8  |
> | + RoPE++$_\text{EC}$ | 42.7  | 38.6  | 33.4  | 21.7  |**34.1**| 32.4  | 29.9  | 24.4  | 24.5  | 18.6  |**26.0**|
> | RoPE + PI            | 37.8  | 34.4  | 30.5  | 13.4  | 29.0  | 15.3  | 16.9  | 12.7  | 11.8  | 9.3   | 13.2  |
> | + RoPE++$_\text{EH}$ | 37.9  | 35.0  | 27.5  | 14.6  | 28.8  | 21.0  | 22.4  | 17.1  | 13.7  | 11.1  |**17.1**|
> | + RoPE++$_\text{EC}$ | 43.0  | 38.7  | 28.8  | 13.6  |**31.0**| 25.7  | 23.4  | 16.4  | 9.4   | 8.0   | 16.6  |
> | RoPE + YaRN          | 37.6  | 35.0  | 33.9  | 27.5  | 33.5  | 26.9  | 25.6  | 19.5  | 16.4  | 12.2  | 20.1  |
> | + RoPE++$_\text{EH}$ | 37.9  | 34.9  | 32.2  | 26.1  | 32.8  | 28.0  | 23.9  | 18.6  | 17.8  | 11.7  | 20.0  |
> | + RoPE++$_\text{EC}$ | 42.9  | 36.5  | 36.3  | 22.2  |**34.4**| 26.3  | 24.1  | 21.1  | 19.8  | 16.9  |**21.6**|
> |***376M Long***       |       |       |       |       |       |       |       |       |       |       |       |
> | RoPE + NTK           | 31.6  | 25.6  | 22.0  | 9.5   | 22.2  | 17.7  | 16.1  | 9.1   | 9.4   | 5.9   | 11.6  |
> | + RoPE++$_\text{EH}$ | 29.9  | 28.4  | 17.6  | 9.4   | 21.3  | 14.1  | 15.6  | 12.2  | 9.9   | 8.3   | 12.0  |
> | + RoPE++$_\text{EC}$ | 36.1  | 33.0  | 29.1  | 17.7  |**29.0**| 19.8  | 19.8  | 16.1  | 15.8  | 12.3  |**16.8**|
> | RoPE + PI            | 36.5  | 33.6  | 19.7  | 10.6  | 25.1  | 19.3  | 12.3  | 10.2  | 10.9  | 10.9  | 12.7  |
> | + RoPE++$_\text{EH}$ | 28.0  | 27.6  | 15.8  | 6.9   | 19.6  | 13.3  | 12.4  | 12.8  | 8.9   | 10.4  | 11.6  |
> | + RoPE++$_\text{EC}$ | 37.0  | 32.4  | 28.3  | 10.6  |**27.1**| 24.0  | 20.7  | 15.9  | 14.3  | 12.3  |**17.4**|
> | RoPE + YaRN          | 36.4  | 32.9  | 28.4  | 15.0  | 28.2  | 22.4  | 16.4  | 11.4  | 10.7  | 11.1  | 14.4  |
> | + RoPE++$_\text{EH}$ | 32.7  | 30.2  | 24.9  | 10.7  | 24.7  | 8.7   | 9.3   | 12.1  | 11.3  | 10.9  | 10.5  |
> | + RoPE++$_\text{EC}$ | 36.0  | 33.9  | 31.7  | 17.8  |**29.8**| 27.4  | 23.6  | 18.0  | 16.9  | 12.3  |**19.6**|

---

> ### Author Response · Authors · 2025-11-25
> **Reply (3/3)**
>
> |                      | Wiki  | LMD   | TQA   | PIQA  | Hella | Wino  | ARC-e | GPQA  | SIQA  | OBQA  | SG    | Avg   |
> |:---------------------|:-----:|:-----:|:-----:|:-----:|:-----:|:-----:|:-----:|:-----:|:-----:|:-----:|:-----:|:-----:|
> |                      | ppl   | ppl   | acc   | acc   | acc   | acc   | acc   | acc   | acc   | acc   | acc   |       |
> |***776M Long***       |       |       |       |       |       |       |       |       |       |       |       |
> | RoPE + NTK           | 14.6  | 27.3  | 35.1  | 68.9  | 43.1  | 51.5  | 47.6  | 21.7  | 40.7  | 20.2  | 42.6  | 41.3  |
> | + RoPE++$_\text{EH}$ | 15.3  | 28.1  | 35.4  | 69.9  | 41.9  | 52.6  | 43.2  | 28.3  | 41.0  | 22.2  | 43.4  | 42.0  |
> | + RoPE++$_\text{EC}$ |**14.4**|**27.1**| 35.2  | 70.4  | 43.7  | 52.6  | 44.8  | 31.8  | 40.8  | 27.6  | 44.3  |**43.5**|
> | RoPE + PI            |**27.8**|**14.8**| 35.8  | 65.4  | 33.8  | 53.1  | 38.8  | 25.3  | 39.6  | 27.2  | 44.7  | 40.4  |
> | + RoPE++$_\text{EH}$ | 28.8  | 15.6  | 37.0  | 66.0  | 33.1  | 52.6  | 40.2  | 25.8  | 38.1  | 26.4  | 44.0  | 40.4  |
> | + RoPE++$_\text{EC}$ |**27.8**| 14.9  | 37.0  | 65.4  | 33.9  | 51.7  | 39.5  | 23.2  | 39.4  | 28.4  | 45.8  |**40.5**|
> | RoPE + YaRN          |**27.3**|**14.5**| 36.7  | 65.6  | 33.9  | 51.1  | 37.7  | 29.3  | 39.7  | 27.6  | 46.3  | 40.9  |
> | + RoPE++$_\text{EH}$ | 28.3  | 15.1  | 38.0  | 66.1  | 33.6  | 51.6  | 40.9  | 26.3  | 39.3  | 26.0  | 43.9  | 40.6  |
> | + RoPE++$_\text{EC}$ |**27.3**|**14.5**| 37.1  | 65.8  | 34.8  | 53.4  | 41.8  | 27.8  | 40.1  | 27.2  | 45.1  |**41.5**|
> |***376M Long***       |       |       |       |       |       |       |       |       |       |       |       |
> | RoPE + NTK           | 20.4  |**33.8**| 35.4  | 64.9  | 34.1  | 50.6  | 40.4  | 21.2  | 39.4  | 27.4  | 43.5  | 39.6  |
> | + RoPE++$_\text{EH}$ | 21.7  | 34.8  | 35.2  | 64.5  | 34.3  | 49.9  | 41.5  | 22.7  | 40.0  | 27.0  | 43.1  | 39.8  |
> | + RoPE++$_\text{EC}$ |**20.0**| 33.9  | 37.1  | 66.1  | 34.1  | 53.4  | 38.1  | 21.2  | 39.2  | 28.4  | 43.7  |**40.1**|
> | RoPE + PI            |**33.4**| 20.1  | 34.9  | 69.4  | 43.1  | 52.5  | 42.9  | 29.3  | 40.0  | 21.6  | 44.3  | 42.0  |
> | + RoPE++$_\text{EH}$ | 34.7  | 20.7  | 35.5  | 68.4  | 41.1  | 53.7  | 44.6  | 22.7  | 40.0  | 26.0  | 42.9  | 41.7  |
> | + RoPE++$_\text{EC}$ | 33.7  |**19.4**| 36.6  | 70.1  | 43.1  | 51.9  | 44.8  | 27.3  | 40.7  | 27.0  | 44.1  |**42.8**|
> | RoPE + YaRN          |**32.8**| 19.3  | 36.1  | 69.9  | 43.5  | 52.6  | 44.1  | 25.3  | 41.2  | 24.0  | 43.4  | 42.2  |
> | + RoPE++$_\text{EH}$ | 33.9  | 20.1  | 35.7  | 68.8  | 41.8  | 55.0  | 45.2  | 25.8  | 40.4  | 24.6  | 42.2  | 42.2  |
> | + RoPE++$_\text{EC}$ | 32.9  |**18.8**| 36.7  | 69.5  | 43.6  | 52.6  | 44.1  | 29.3  | 41.0  | 30.2  | 44.0  |**43.4**|
>
> [1] HoPE: Hyperbolic Rotary Positional Encoding for Stable Long-Range Dependency Modeling in Large Language Models https://www.arxiv.org/abs/2509.05218
>
> [2] DoPE: Denoising Rotary Position Embedding https://arxiv.org/abs/2511.09146
>
> [3] Pythia: A Suite for Analyzing Large Language Models Across Training and Scaling https://github.com/EleutherAI/pythia
>
> [4] Train Short, Test Long: Attention with Linear Biases Enables Input Length Extrapolation https://arxiv.org/abs/2108.12409
>
> [5] Beyond Homogeneous Attention: Memory-Efficient LLMs via Fourier-Approximated KV Cache https://arxiv.org/abs/2506.11886
>
> [6] DuoAttention: Efficient Long-Context LLM Inference with Retrieval and Streaming Heads https://arxiv.org/abs/2410.10819
>
> [7] DeepSeek-V2: A Strong, Economical, and Efficient Mixture-of-Experts Language Model https://arxiv.org/abs/2405.04434
>
> [8] GQA: Training Generalized Multi-Query Transformer Models from Multi-Head Checkpoints https://arxiv.org/abs/2305.13245
>
> [9] MInference 1.0: Accelerating Pre-filling for Long-Context LLMs via Dynamic Sparse Attention https://arxiv.org/abs/2407.02490
>
> [10] Extending Context Window of Large Language Models via Positional Interpolation https://arxiv.org/abs/2306.15595
>
> [11] YaRN: Efficient Context Window Extension of Large Language Models https://arxiv.org/abs/2309.00071

---

### Official Review · Reviewer_RADi · 2025-10-31

**Soundness:** 3
**Presentation:** 3
**Contribution:** 3
**Rating:** 6
**Confidence:** 2

**Summary:**

The paper proposes RoPE++, an extension of Rotary Position Embeddings (RoPE) designed to enhance the modeling of long-context dependencies in Large Language Models (LLMs) utilizing imaginary component of RoPE. It analyses how it affects cache and parameters efficiency, impacts length extrapolation.

**Strengths:**

- proposes new method for positional encoding which shows promise, especially for long context
- performs extensive experimentations on different datasets
- includes several positional encoding methods as baseline

**Weaknesses:**

- would be great to have baseline for  RoPE method, which target long context
- theoretical justification of method could be improved

**Questions:**

Could you please clarify the changes (if any)  in the number of parameters, activations and FLOPs (estimate) for RoPE++ methods? Am I understanding correctly that Wq parameters are shared between imaginary and real heads? What about Wo size?

Considering that RoPE++ performs well on long-context task, would be great to have baseline with one of the derivate RoPE methods, which target long context (i.e one of Linear Position Interpolation (PI), NTK Scaling, YARN, LongRoPE, p-RoPE)

In section 5.1, is analysing inference time only? Could you add some information how RoPE++ affects training?

line 193:  "long context decay". Could you please clarify what distribution of queries and keys you are considering?  As shown in https://arxiv.org/abs/2410.06205v1 decay is clear if RoPE is applied to constant queries and keys, but not so much if queries and keys are Gaussian.

nit: line 256.  "share the same parameter" same parameters?

---

> ### Author Response · Authors · 2025-11-25
> **Reply (1/3)**
>
> First, we sincerely thank the reviewer for recognizing our promising design, extensive experiments, and rich comparison.
>
> **Regarding W1 and Q2 on the derivative RoPE methods, which target long context**
>
> Thank you for this constructive review. RoPE++ can not only be combined with NTK for context extension during long-context training, but can also be combined with Linear PI[1] and YaRN[2]. Across 376M and 776M model sizes, we conduct extensive experiments, with the interpolation coefficient $s = 8$ for Linear PI and $s = 32$ for YaRN. RoPE++ consistently achieves the highest composite scores on RULER, BABILong, and the average score of short-context tasks, confirming its advantage and generalization. These results will be added to the revised paper.
>
> |                      | RU    | LE    | R     |       |       | BA    | BI    | Lo    | ng    |       |       |
> |:---------------------|:-----:|:-----:|:-----:|:-----:|:-----:|:-----:|:-----:|:-----:|:-----:|:-----:|:-----:|
> |                      | 4k    | 8k    | 16k   | 32k   | Avg   | 2k    | 4k    | 8k    | 16k   | 32k   | Avg   |
> |***776M Long***       |       |       |       |       |       |       |       |       |       |       |       |
> | RoPE + NTK           | 37.4  | 35.1  | 33.0  | 21.2  | 31.7  | 33.5  | 30.7  | 23.6  | 22.0  | 15.1  | 25.0  |
> | + RoPE++$_\text{EH}$ | 38.7  | 35.4  | 33.8  | 24.6  | 33.1  | 31.9  | 26.5  | 18.6  | 16.2  | 11.0  | 20.8  |
> | + RoPE++$_\text{EC}$ | 42.7  | 38.6  | 33.4  | 21.7  |**34.1**| 32.4  | 29.9  | 24.4  | 24.5  | 18.6  |**26.0**|
> | RoPE + PI            | 37.8  | 34.4  | 30.5  | 13.4  | 29.0  | 15.3  | 16.9  | 12.7  | 11.8  | 9.3   | 13.2  |
> | + RoPE++$_\text{EH}$ | 37.9  | 35.0  | 27.5  | 14.6  | 28.8  | 21.0  | 22.4  | 17.1  | 13.7  | 11.1  |**17.1**|
> | + RoPE++$_\text{EC}$ | 43.0  | 38.7  | 28.8  | 13.6  |**31.0**| 25.7  | 23.4  | 16.4  | 9.4   | 8.0   | 16.6  |
> | RoPE + YaRN          | 37.6  | 35.0  | 33.9  | 27.5  | 33.5  | 26.9  | 25.6  | 19.5  | 16.4  | 12.2  | 20.1  |
> | + RoPE++$_\text{EH}$ | 37.9  | 34.9  | 32.2  | 26.1  | 32.8  | 28.0  | 23.9  | 18.6  | 17.8  | 11.7  | 20.0  |
> | + RoPE++$_\text{EC}$ | 42.9  | 36.5  | 36.3  | 22.2  |**34.4**| 26.3  | 24.1  | 21.1  | 19.8  | 16.9  |**21.6**|
> |***376M Long***       |       |       |       |       |       |       |       |       |       |       |       |
> | RoPE + NTK           | 31.6  | 25.6  | 22.0  | 9.5   | 22.2  | 17.7  | 16.1  | 9.1   | 9.4   | 5.9   | 11.6  |
> | + RoPE++$_\text{EH}$ | 29.9  | 28.4  | 17.6  | 9.4   | 21.3  | 14.1  | 15.6  | 12.2  | 9.9   | 8.3   | 12.0  |
> | + RoPE++$_\text{EC}$ | 36.1  | 33.0  | 29.1  | 17.7  |**29.0**| 19.8  | 19.8  | 16.1  | 15.8  | 12.3  |**16.8**|
> | RoPE + PI            | 36.5  | 33.6  | 19.7  | 10.6  | 25.1  | 19.3  | 12.3  | 10.2  | 10.9  | 10.9  | 12.7  |
> | + RoPE++$_\text{EH}$ | 28.0  | 27.6  | 15.8  | 6.9   | 19.6  | 13.3  | 12.4  | 12.8  | 8.9   | 10.4  | 11.6  |
> | + RoPE++$_\text{EC}$ | 37.0  | 32.4  | 28.3  | 10.6  |**27.1**| 24.0  | 20.7  | 15.9  | 14.3  | 12.3  |**17.4**|
> | RoPE + YaRN          | 36.4  | 32.9  | 28.4  | 15.0  | 28.2  | 22.4  | 16.4  | 11.4  | 10.7  | 11.1  | 14.4  |
> | + RoPE++$_\text{EH}$ | 32.7  | 30.2  | 24.9  | 10.7  | 24.7  | 8.7   | 9.3   | 12.1  | 11.3  | 10.9  | 10.5  |
> | + RoPE++$_\text{EC}$ | 36.0  | 33.9  | 31.7  | 17.8  |**29.8**| 27.4  | 23.6  | 18.0  | 16.9  | 12.3  |**19.6**|

---

> ### Author Response · Authors · 2025-11-25
> **Reply (2/3)**
>
> |                      | Wiki  | LMD   | TQA   | PIQA  | Hella | Wino  | ARC-e | GPQA  | SIQA  | OBQA  | SG    | Avg   |
> |:---------------------|:-----:|:-----:|:-----:|:-----:|:-----:|:-----:|:-----:|:-----:|:-----:|:-----:|:-----:|:-----:|
> |                      | ppl   | ppl   | acc   | acc   | acc   | acc   | acc   | acc   | acc   | acc   | acc   |       |
> |***776M Long***       |       |       |       |       |       |       |       |       |       |       |       |
> | RoPE + NTK           | 14.6  | 27.3  | 35.1  | 68.9  | 43.1  | 51.5  | 47.6  | 21.7  | 40.7  | 20.2  | 42.6  | 41.3  |
> | + RoPE++$_\text{EH}$ | 15.3  | 28.1  | 35.4  | 69.9  | 41.9  | 52.6  | 43.2  | 28.3  | 41.0  | 22.2  | 43.4  | 42.0  |
> | + RoPE++$_\text{EC}$ |**14.4**|**27.1**| 35.2  | 70.4  | 43.7  | 52.6  | 44.8  | 31.8  | 40.8  | 27.6  | 44.3  |**43.5**|
> | RoPE + PI            |**27.8**|**14.8**| 35.8  | 65.4  | 33.8  | 53.1  | 38.8  | 25.3  | 39.6  | 27.2  | 44.7  | 40.4  |
> | + RoPE++$_\text{EH}$ | 28.8  | 15.6  | 37.0  | 66.0  | 33.1  | 52.6  | 40.2  | 25.8  | 38.1  | 26.4  | 44.0  | 40.4  |
> | + RoPE++$_\text{EC}$ |**27.8**| 14.9  | 37.0  | 65.4  | 33.9  | 51.7  | 39.5  | 23.2  | 39.4  | 28.4  | 45.8  |**40.5**|
> | RoPE + YaRN          |**27.3**|**14.5**| 36.7  | 65.6  | 33.9  | 51.1  | 37.7  | 29.3  | 39.7  | 27.6  | 46.3  | 40.9  |
> | + RoPE++$_\text{EH}$ | 28.3  | 15.1  | 38.0  | 66.1  | 33.6  | 51.6  | 40.9  | 26.3  | 39.3  | 26.0  | 43.9  | 40.6  |
> | + RoPE++$_\text{EC}$ |**27.3**|**14.5**| 37.1  | 65.8  | 34.8  | 53.4  | 41.8  | 27.8  | 40.1  | 27.2  | 45.1  |**41.5**|
> |***376M Long***       |       |       |       |       |       |       |       |       |       |       |       |
> | RoPE + NTK           | 20.4  |**33.8**| 35.4  | 64.9  | 34.1  | 50.6  | 40.4  | 21.2  | 39.4  | 27.4  | 43.5  | 39.6  |
> | + RoPE++$_\text{EH}$ | 21.7  | 34.8  | 35.2  | 64.5  | 34.3  | 49.9  | 41.5  | 22.7  | 40.0  | 27.0  | 43.1  | 39.8  |
> | + RoPE++$_\text{EC}$ |**20.0**| 33.9  | 37.1  | 66.1  | 34.1  | 53.4  | 38.1  | 21.2  | 39.2  | 28.4  | 43.7  |**40.1**|
> | RoPE + PI            |**33.4**| 20.1  | 34.9  | 69.4  | 43.1  | 52.5  | 42.9  | 29.3  | 40.0  | 21.6  | 44.3  | 42.0  |
> | + RoPE++$_\text{EH}$ | 34.7  | 20.7  | 35.5  | 68.4  | 41.1  | 53.7  | 44.6  | 22.7  | 40.0  | 26.0  | 42.9  | 41.7  |
> | + RoPE++$_\text{EC}$ | 33.7  |**19.4**| 36.6  | 70.1  | 43.1  | 51.9  | 44.8  | 27.3  | 40.7  | 27.0  | 44.1  |**42.8**|
> | RoPE + YaRN          |**32.8**| 19.3  | 36.1  | 69.9  | 43.5  | 52.6  | 44.1  | 25.3  | 41.2  | 24.0  | 43.4  | 42.2  |
> | + RoPE++$_\text{EH}$ | 33.9  | 20.1  | 35.7  | 68.8  | 41.8  | 55.0  | 45.2  | 25.8  | 40.4  | 24.6  | 42.2  | 42.2  |
> | + RoPE++$_\text{EC}$ | 32.9  |**18.8**| 36.7  | 69.5  | 43.6  | 52.6  | 44.1  | 29.3  | 41.0  | 30.2  | 44.0  |**43.4**|
>
> **Regarding W2 on theoretical justification**
>
> We thank the reviewer for raising this critical question. In Section 3.2 and Appendix B, we state the two important properties of RoPE, semantic aggregation and long-context decay. On one hand, when $\bm{q}, \bm{k}$ vectors are semantically close, their attention score remains large on average, regardless of relative distance $\Delta{t}$. Mathematically, if we have a vector $\bm{k}$ that is independent and identically distributed with respect to $\bm{q}$, with average $\mu$ and variance $\sigma^2$ for every feature dimension, and a vector that is only slightly perturbed with respect to $\bm{q}+\bm{\varepsilon}$, the expected attention score difference can be calculated as follows and proved to be positive. That is to say,
>
> $$\mathbb{E}_{\bm{q},\bm{k},\bm{\varepsilon}}\left[\bm{q}^\top \bm{\mathcal{R}}_{-\Delta{t}}(\bm{q}+\bm{\varepsilon}) - \bm{q}^\top \bm{\mathcal{R}}_{-\Delta{t}} \bm{k}\right]>0$$
>
> On the other hand, as $\Delta{t}$ increases, the attention between any $\bm{q}, \bm{k}$ decreases on average. We can similarly derive this by showing that the expectation of the attention score $\mathbb{E}_{\bm{q},\bm{k}}\left[\bm{q}^\top \bm{\mathcal{R}}_{-\Delta{t}}\bm{k}\right]$, is almost monotonically decaying with the increase of $\Delta{t}$. Based on the detailed derivation in Appendix B (page 15), both properties are reduced to a positive, decaying cosine integral function.
>
> $$\tilde{c}_\text{Re}(\Delta{t})=\int\limits_{{10}^{-4}}^{1}\dfrac{\cos{\theta{t}}}{\theta\ln{10}^4}\mathrm{d}\theta = \mathrm{Ci}(\Delta{t})-\mathrm{Ci}\left(\frac{\Delta{t}}{{10}^{4}}\right)$$

---

> ### Author Response · Authors · 2025-11-25
> **Reply (3/3)**
>
> The detailed derivations are detailed in Appendix B on page 15. These derivations target the original RoPE, i.e., the real-part result of the complex representation in RoPE++.
>
> $$\bm{A}_{t,s}^\text{Re}=\mathrm{Re}\left[\sum_{n=0}^{d/2-1}{\tilde{q}_t^{(n)}\tilde{k}_s^{(n)*}e^{i\theta_n(s-t)}}\right]=\left(\bm{\mathcal{R}}_{\Theta,t}\bm{q}_t\right)^\top\bm{\mathcal{R}}_{\Theta,s}\bm{k}_s=\bm{q}_t^\top\bm{\mathcal{R}}_{\Theta,s-t}\bm{k}_s$$
>
> For the imaginary component lost in RoPE’s complex representation, we derive a sine integral function.
>
> $$\tilde{c}_\text{Im}=\int\limits_{{10}^{-4}}^{1}\dfrac{\sin{\theta{t}}}{\theta\ln{10}^4}\mathrm{d}\theta = \mathrm{Si}(\Delta{t})-\mathrm{Si}\left(\frac{\Delta{t}}{{10}^{4}}\right)$$
>
> **Regarding Q1 and Q3 on architectural clarification**
>
> Thank you for this critical question. We apologize for the lack of sufficient clarification in the paper. In RoPE++, both RoPE++$\_\text{EH}$ and RoPE++$\_\text{EC}$, the query projections Wq are indeed shared between the real and imaginary attention, exactly as you state. Because the imaginary attention doubles the number of output heads, Wo must be twice the size of Wq. Under the RoPE++$\_\text{EH}$ setting, Wo equals the original RoPE size, whereas under the RoPE++$\_\text{EC}$ setting, Wo is twice as large as that in vanilla RoPE. We now measure the throughput (TGS, tokens per GPU per second) for the 776M and 376M models at 4k and 32k training context lengths, as well as the final parameter storage as follows. These details will be added to our paper.
>
> This function is also non-negative, satisfying semantic aggregation, but decays slowly beyond a certain relative distance, implying that the model can capture long-context semantic dependencies more effectively. We therefore reintroduce the imaginary-part computation into the attention calculation as an additional attention head, yielding our method RoPE++.
>
> |                    |  TGS-4k   | TGS-32k   | Storage |
> |:-------------------|:---------:|:---------:|:-------:|
> | 776M               |           |           |         |
> | RoPE               | 49617.16  | 29019.63  | 1.6G    |
> | RoPE++$_\text{EH}$ | 50574.41  | 29399.33  | 1.5G    |
> | RoPE++$_\text{EC}$ | 44431.19  | 22631.14  | 1.6G    |
> | 376M               |           |           |         |
> | RoPE               | 80248.16  | 53317.42  | 782M    |
> | RoPE++$_\text{EH}$ | 80248.16  | 53498.78  | 766M    |
> | RoPE++$_\text{EC}$ | 70457.45  | 37271.66  | 798M    |
>
> **Regarding Q4 on long context decay**
>
> Thank you for this insightful question. We are happy that the reviewer has a deep understanding of positional embeddings. We do not assume any specific distribution of the $\bm{q},\bm{k}$ vectors. The only premise is that vector k is i.i.d. with respect to q, with mean $\mu$ and variance $\sigma^2$ for every feature dimension. Unlike p-RoPE’s case-study-level discussion of long-context decay [3], our analysis is expectation-based, i.e., the average fluctuation of the real or imaginary attention, as detailed in the reply to W2. The reviewer’s remark on p-RoPE’s long-context decay is orthogonal to our contribution. We can likewise drop the rotary angles for the low-frequency dimensions in both the real and imaginary branches of RoPE++. Since there are already many verification experiments, we will add experiments later to verify that RoPE++ retains its advantage over vanilla RoPE when combined with p-RoPE refinements.
>
> We also thank the reviewer for the careful reading that raises typos. The above analyses and experimental results will be incorporated into the revised version of our paper. We are grateful once again for these suggestions.
>
> [1] Extending Context Window of Large Language Models via Positional Interpolation https://arxiv.org/abs/2306.15595
>
> [2] YaRN: Efficient Context Window Extension of Large Language Models https://arxiv.org/abs/2309.00071
>
> [3] Round and Round We Go! What makes Rotary Positional Encodings useful? https://arxiv.org/abs/2410.06205

---

### Official Review · Reviewer_ADoG · 2025-11-01

**Soundness:** 3
**Presentation:** 2
**Contribution:** 2
**Rating:** 4
**Confidence:** 3

**Summary:**

The paper proposes to extend RoPE by including the imaginary component when interpreting the computation of attention weights as multiplications on complex vectors. The claims are mainly the following:
- "... the negative imaginary part of attention, shows that, compared with the real attention exhibiting stronger semantic locality, the imaginary heads attend more to long-context information as shown in Figure 1, promising gains on long-context tasks"
- RoPE++ outperforms RoPE in the ablations with models of sizes 776M and 376M.
- one of the variant of RoPE++ reduces the KV cache by half but delivers on par performance.

**Strengths:**

1. The idea is novel. The authors had the good observation that RoPE only captures the real part when interpreted as complex multiplications.
2. Good theoretic motivations and explanations.
3. The evaluation uses a good set of benchmarks and looks convincing given their training horizon (50B tokens) and their scale, though I'm skeptical how good it will continue to be if we train for longer for reasonable amount of tokens.

**Weaknesses:**

1. Experiment setup doesn't seem to have enough scale to convincingly show the gain in model pre-training. 50B token is unfortunately sometimes too small to have confidence about certain pre-training signals, though I understand that there is typically a budget issue in academia.
2. I'm skeptical about whether the setup is bug-free and uses reasonable hyperparameters, as the training for ALiBi and NoPE seem to have to take compromises.

**Questions:**

1. In the experiment section, you mentioned "Since ALiBi and NoPE train unstably at 4k, as we have tried, we train
them on 1k context length while keeping the batch size the same.". This made me very skeptical about whether the parameters you fix is reasonable, and your experiment setup is correct. In my experiences, with reasonable effort, all these positional encoding can be trained without problems up to tens of billions of parameters for trillions of tokens. Have you carefully debugged your experiments? Have you tracked some basic training health metrics across layers? Have you considered redoing and comparing with some reproducible pre-training recipes such as OLMo/Pythia/... to ensure no regression due to bugs and mistakes? Have you sweeped hyperparameters for the architectures you chose?
2. Do you have experiment to support the "real part having better locality and imaginary part having better dependency capture", rather than the hand-wavy point-wise argument by only looking at magnitudes? This could become a very interesting data point, regardless whether the experiment supports or rejects the claim.

---

> ### Author Response · Authors · 2025-11-25
> **Reply (1/3)**
>
> First, we sincerely thank the reviewer for recognizing our novel idea, theoretical motivations, as well as explanations, and comprehensive evaluation.
>
> **Regarding W1 on data scale**
>
> Thank you for this constructive review. We acknowledge that training on more data is constructive to improve scalability, although concurrent or earlier work still only performs pre-training validation on model sizes smaller than 1B and data volumes smaller than 50B tokens [1-3]. Based on the experiments on 776M and 376M, 50B tokens are already sufficient for convergence at this scale, as evidenced by plateaued training loss, validation loss, and average short-context scores. We verify this judgement by evaluating 776M checkpoints pre-trained at 50B, 60B, and 70B tokens while the learning rate remains constant. Beyond 50B tokens, the model shows no significant further gain. Note that these 776M results at 50B tokens are obtained without learning-rate annealing, so they differ slightly from the scores reported in the paper.
>
> |***376M***            | 5B      | 10B     | 15B     | 20B     | 30B     | 40B     | 50B     |
> |:---------------------|:-------:|:-------:|:-------:|:-------:|:-------:|:-------:|:-------:|
> |*Training Loss*       |         |         |         |         |         |         |         |
> | RoPE                 | 3.4698  | 3.3576  | 3.2249  | 3.3459  | 3.2945  | 3.2946  | 3.1665  |
> | + RoPE++$_\text{EH}$ | 3.4904  | 3.3821  | 3.2511  | 3.3708  | 3.3202  | 3.3194  | 3.1931  |
> | + RoPE++$_\text{EC}$ | 3.4567  | 3.3473  | 3.2203  | 3.3373  | 3.2848  | 3.2904  | 3.1612  |
> |*Validation Loss*     |         |         |         |         |         |         |         |
> | RoPE                 | 3.5509  | 3.4358  | 3.3933  | 3.3629  | 3.3299  | 3.3177  | 3.1881  |
> | + RoPE++$_\text{EH}$ | 3.5772  | 3.4618  | 3.4217  | 3.3907  | 3.3567  | 3.3430  | 3.2141  |
> | + RoPE++$_\text{EC}$ | 3.5362  | 3.4254  | 3.3870  | 3.3569  | 3.3251  | 3.3140  | 3.2683  |
> |*Average Score*       |         |         |         |         |         |         |         |
> | RoPE                 |**39.2** |**39.6** |**40.4** |**39.9** | 39.5    | 39.8    | 40.1    |
> | + RoPE++$_\text{EH}$ | 38.2    |**39.6** | 40.3    | 38.9    | 39.5    | 39.8    | 40.3    |
> | + RoPE++$_\text{EC}$ | 38.8    | 38.8    | 39.3    | 39.6    |**40.3** |**40.2** |**41.0** |
>
> |***776M***            | 5B      | 10B     | 15B     | 20B     | 30B     | 40B     | 50B*    | 60B     | 70B     |
> |:---------------------|:-------:|:-------:|:-------:|:-------:|:-------:|:-------:|:-------:|:-------:|:-------:|
> |*Training Loss*       |         |         |         |         |         |         |         |         |         |
> | RoPE                 | 3.2826  | 3.1621  | 3.0279  | 3.1418  | 3.0889  | 3.0913  | 3.0502  | 3.0208  | 3.0307  |
> | + RoPE++$_\text{EH}$ | 3.3104  | 3.1955  | 3.0622  | 3.1701  | 3.1236  | 3.1237  | 3.0798  | 3.0496  | 3.0596  |
> | + RoPE++$_\text{EC}$ | 3.2797  | 3.1620  | 3.0294  | 3.1377  | 3.0890  | 3.0889  | 3.0489  | 3.0200  | 3.0294  |
> |*Validation Loss*     |         |         |         |         |         |         |         |         |         |
> | RoPE                 | 3.3480  | 3.2279  | 3.1842  | 3.1552  | 3.1214  | 3.1083  | 3.0912  | 3.0817  | 3.0751  |
> | + RoPE++$_\text{EH}$ | 3.3865  | 3.2648  | 3.2184  | 3.1877  | 3.1545  | 3.1401  | 3.1232  | 3.1168  | 3.1050  |
> | + RoPE++$_\text{EC}$ | 3.3541  | 3.2305  | 3.1872  | 3.1550  | 3.1205  | 3.1061  | 3.0903  | 3.0816  | 3.0733  |
> |*Average Score*       |         |         |         |         |         |         |         |         |         |
> | RoPE                 | 39.3    | 40.8    |**41.3** |**41.5** | 41.4    | 41.5    |**41.7** |**42.3** | 41.8    |
> | + RoPE++$_\text{EH}$ |**40.0** |**40.9** | 41.2    | 40.6    | 41.7    |**42.2** | 41.3    | 41.9    |**42.1** |
> | + RoPE++$_\text{EC}$ | 38.7    | 40.4    | 41.1    | 41.0    |**41.8** | 41.7    |**41.7** | 41.8    | 41.9    |

---

> ### Author Response · Authors · 2025-11-25
> **Reply (2/3)**
>
> Therefore, we have pre-trained a larger 1.5B model under the same training parameters used for 776M/376M and evaluated it at 50B tokens in 4k context length, followed by 5B tokens in 32k context length. RoPE++ again achieves the best average scores on RULER, BABILong, and short-context tasks. To sum up, within the limits of our available computing resources, we have successfully validated the effectiveness of RoPE++ on all three model scales. We are continuing to train this 1.5B model to 100B tokens. Given limited resources and time, we will update the paper with the final results later. We sincerely appreciate the reviewer’s constructive feedback.
>
> |                      | Wiki  | LMD   | TQA   | PIQA  | Hella | Wino  | ARC-e | GPQA  | SIQA  | OBQA  | SG    | Avg   |
> |:---------------------|:-----:|:-----:|:-----:|:-----:|:-----:|:-----:|:-----:|:-----:|:-----:|:-----:|:-----:|:-----:|
> | ***1.5B Short***     | ppl   | ppl   | acc   | acc   | acc   | acc   | acc   | acc   | acc   | acc   | acc   |       |
> | RoPE                 | 27.9  |**15.3**| 36.4  | 70.8  | 46.2  | 53.2  | 44.8  | 25.8  | 39.4  | 24.6  | 44.8  | 42.9  |
> | + RoPE++$_\text{EH}$ | 28.0  | 16.0  | 37.1  | 71.2  | 45.6  | 53.6  | 45.9  | 25.8  | 40.8  | 27.8  | 44.3  |**43.6**|
> | + RoPE++$_\text{EC}$ |**27.4**| 15.5  | 35.1  | 69.5  | 46.3  | 53.3  | 43.6  | 28.3  | 40.5  | 23.6  | 45.6  | 42.9  |
>
> |                      | RU    | LE    | R     |       |       |       | BA    | BI    | Lo    | ng    |       |       |       |
> |:---------------------|:-----:|:-----:|:-----:|:-----:|:-----:|:-----:|:-----:|:-----:|:-----:|:-----:|:-----:|:-----:|:-----:|
> | ***1.5B Long***      | 4k    | 8k    | 16k   | 32k   | 64k   | Avg   | 2k    | 4k    | 8k    | 16k   | 32k   | 64k   | Avg   |
> | RoPE                 | 50.5  | 39.9  | 34.6  |**31.9**| 18.3  | 35.1  | 25.9  | 29.7  |**31.8**|**30.5**|**18.5**| 13.5  | 29.5  |
> | + RoPE++$_\text{EH}$ | 42.5  | 38.1  | 33.1  | 28.7  | 12.7  | 31.0  |**40.0**|**35.9**| 30.5  | 25.1  | 14.7  |**15.8**|**32.9**|
> | + RoPE++$_\text{EC}$ |**53.1**|**45.7**|**39.0**| 30.6  |**18.9**|**37.5**| 25.1  | 28.3  | 21.4  | 16.6  | 12.6  | 12.3  | 22.9  |
>
> **Regarding W2 and Q1 on comparison with ALiBi and NoPE**
>
> Thank you for these critical questions and for pushing us to ensure the validity, robustness, and reproducibility of our experimental design. Following your guidance, we have revisited our entire experimental setup and conducted a new set of experiments to address your concerns directly:
> 1. **ALiBi: Re-training at 4k.** Thank you for your valuable feedback. Following your suggestion, we conducted an in-depth review of our initial ALiBi implementation, identifying and correcting an implementation issue. To ensure the correctness of our code, we meticulously aligned our implementation with an established open-source model, Baichuan2-13B[4], which also employs a 4k context length with ALiBi. After this alignment, our model successfully demonstrated stable training at the 4k context length. The new training run uses a configuration identical to that of our proposed method and other baselines, ensuring a fair and correct comparison.
> 2. **NoPE: Adopting a Reproducible Baseline Recipe (Pythia).** Your suggestion to benchmark against reproducible pre-training recipes like Pythia is a good point. Our initial choice of a 1k context length for the NoPE baseline was due to significant performance degradation we observed when attempting to train it directly at a 4k context length. This finding aligns with the fact that, to our knowledge, there are no established reports of mainstream fully attention-based LLMs being successfully trained from scratch using NoPE at such context lengths. To address this and your suggestion simultaneously, we have now replaced NoPE with Partial RoPE (as used in Pythia). We trained this new, more stable baseline at a 4k context length, directly anchoring our comparison to a community-trusted standard as you recommended.

---

> ### Author Response · Authors · 2025-11-25
> **Reply (3/3)**
>
> During this new round of experiments, we took extra care to monitor key training health metrics, including loss curves and gradient norms across layers, to ensure the stability and correctness of the training process for all models. This diligent process confirmed that the new 4k ALiBi and Partial RoPE models trained stably to convergence. The updated evaluation results of the 376M and 776M model under this fair comparison framework are presented in the table below.
>
> | | Wiki|LMD|TQA|PIQA|Hella|Wino|ARC-e|GPQA|SIQA|OBQA|SG|Avg|
> |:-|:-:|:-:|:-:|:-:|:-:|:-:|:-:|:-:|:-:|:-:|:-:|:-:|
> ||ppl|ppl|acc|acc|acc|acc|acc|acc|acc|acc|acc||
> |***776M Short***| | | | | | | | | | | | |
> | RoPE|14.8|27.3|35.5|70.1|43.7|52.3|43.4|25.8|41.3|21.8|43.6|42.0|
> | RoPE++$_\text{EH}$|15.6|28.1|35.4|69.6|42.7|53.5|45.0|25.8|41.6|26.8|42.4|42.5|
> | RoPE++$_\text{EC}$|14.8|27.3|36.1|69.3|43.6|52.3|43.7|28.3|40.1|27.6|44.4|**42.8**|
> | FoPE|**14.7**|27.1|33.6|68.7|43.4|52.9|45.0|24.8|39.7|24.8|45.4|42.0|
> | Pythia|14.8|**26.9**| 35.8|68.8|42.9|52.1|39.5|22.2|42.0|21.2|43.6|40.9|
> | ALiBi|15.2|28.3|35.2|70.2|43.7|53.6|43.2|23.7|40.6|27.6|45.9|42.6|
> |***376M Short***| | | | | | | | | | | | |
> | RoPE|19.9|32.7|35.5|66.3|34.8|50.9|39.3|24.8|38.6|27.4|43.7|40.1|
> | RoPE++$_\text{EH}$|20.8|33.6|36.3|66.4|34.5|52.5|40.9|23.7|40.5|24.8|43.2|40.3|
> | RoPE++$_\text{EC}$|19.4|**32.6**| 37.3|68.0|35.6|53.0|41.3|25.8|40.3|23.2|44.8|**41.0**|
> | FoPE|**19.3**|33.0|33.8|65.9|34.5|53.0|37.0|28.8|39.5|24.2|43.6|40.0|
> | Pythia|19.2|32.9|34.7|65.8|34.9|51.5|41.3|21.2|39.7|25.6|42.5|39.7|
> | ALiBi|21.2|34.6|33.8|66.1|34.2|51.1|44.4|24.8|38.7|27.4|43.9|40.5|
>
> As the revised results show, after ensuring all baselines are correctly implemented, trained under a fair 4k context, and aligned with reproducible recipes, our RoPE++$_\text{EC}$ continues to hold a performance advantage over ALiBi and the Partial RoPE baselines on downstream tasks. We are confident that these new experiments, conducted with rigorous debugging and adherence to best practices, provide a much stronger and more trustworthy foundation for our claims. Thank you again for your invaluable guidance, which improves the quality of our work. We will carefully revise this part in our paper.
>
> **Regarding Q2 on dependency capture**
>
> Thank you for your question. We have presented an ablation study that analyzes the role of imaginary attention in modeling long-context dependencies in Section 5.2. First, owing to the bias introduced by its characteristic curve, the visualized imaginary heads concentrate more strongly on the initial tokens (Figure 5a–d, f–i), showing a pattern that earlier works [5,6] link to superior long-context semantic capture. Second, to quantify this, we inject Gaussian noise with equal standard deviation into the imaginary and real attention components separately and compare the resulting change in RoPE++ performance, as measured by the average RULER-4k score. The results are presented in Figures 5e and 5j and detailed in the table below. When the standard deviation $\sigma$ is small ($\sigma<0.2$), scores with corrupted real or imaginary attentions stay close to the baseline. When it is large enough ($\sigma=1.5$), both drop sharply. Importantly, in the intermediate range, adding noise to imaginary attention always performs worse than corrupting the real part. Thus, impairing the imaginary heads degrades long-context performance more, confirming that imaginary attention plays a more dominant role in capturing long dependency.
>
> | $\sigma$ | 0.0   | 0.2   | 0.4   | 0.6   | 0.8   | 1.0   | 1.2   | 1.5  |
> |:------------------------|:-----:|:-----:|:-----:|:-----:|:-----:|:-----:|:-----:|:----:|
> |376M RoPE++$_\text{EC}$  |  |  |  |  |  |  |  |  |
> | Real | 31.6  | 31.6  | 31.1  | 29.5  | 25.2  | 17.6  | 10.2  | 3.8  |
> | Imag | 31.6  | 31.1  | 30.0  | 26.2  | 19.4  | 12.8  | 7.4   | 3.3  |
> | Both  | 31.6  | 30.4  | 28.2  | 20.7  | 12.0  | 4.2   | 2.4   | 2.8  |
> | 776M RoPE++$_\text{EC}$ |  |  |  |  |  |  |  |  |
> | Real | 36.6  | 36.6  | 36.9  | 34.5  | 31.8  | 25.4  | 15.7  | 2.1  |
> | Imag | 36.6  | 36.5  | 35.7  | 33.9  | 28.8  | 17.3  | 7.6   | 1.1  |
> | Both | 36.6  | 36.7  | 34.8  | 31.1  | 21.5  | 4.5   | 1.1   | 2.5  |
>
> [1] HoPE: Hyperbolic Rotary Positional Encoding for Stable Long-Range Dependency Modeling in Large Language Models https://arxiv.org/abs/2509.05218
>
> [2] PaTH Attention: Position Encoding via Accumulating Householder Transformations https://arxiv.org/abs/2505.16381
>
> [3] Fourier Position Embedding: Enhancing Attention's Periodic Extension for Length Generalization https://arxiv.org/abs/2412.17739
>
> [4] Baichuan 2: Open Large-scale Language Models https://arxiv.org/abs/2309.10305 https://huggingface.co/baichuan-inc/Baichuan2-13B-Chat
>
> [5] Beyond Homogeneous Attention: Memory-Efficient LLMs via Fourier-Approximated KV Cache https://arxiv.org/abs/2506.11886
>
> [6] DuoAttention: Efficient Long-Context LLM Inference with Retrieval and Streaming Heads https://arxiv.org/abs/2410.10819

---

> ### Author Response · Authors · 2025-12-02
> **Extended Reply**
>
> Regarding data scale, we supplement the 1.5B model’s training loss, validation loss, and average short-context score at 5B, 10B, 15B, 20B, 30B, 40B, 50B, 60B, 70B, 80B, 90B, and 100B tokens. Beyond 50B tokens, the model still shows no significant further gain. We therefore report only the 50B results in the main paper and will provide the full data-scale analysis in the appendix. Thank you for your review.
>
> |***1.5B***            | 5B      | 10B     | 15B     | 20B     | 30B     | 40B     | 50B     | 60B     | 70B     | 80B     | 90B     | 100B    |
> |:---------------------|:-------:|:-------:|:-------:|:-------:|:-------:|:-------:|:-------:|:-------:|:-------:|:-------:|:-------:|:-------:|
> |*Training Loss*       |         |         |         |         |         |         |         |         |         |         |         |         |
> | RoPE                 | 3.1798  | 3.0462  | 2.9492  | 2.9048  | 2.9835  | 2.8967  | 2.9447  | 2.8384  | 2.9855  | 2.8224  | 2.9697  | 2.9775  |
> | + RoPE++$_\text{EH}$ | 3.2051  | 3.0718  | 2.9724  | 2.9333  | 3.0104  | 2.9244  | 2.9688  | 2.8619  | 3.0110  | 2.8491  | 2.9980  | 3.0018  |
> | + RoPE++$_\text{EC}$ | 3.1708  | 3.0384  | 2.9418  | 2.9025  | 2.9824  | 2.8916  | 2.9404  | 2.8307  | 2.9775  | 2.8177  | 2.9664  | 2.9736  |
> |*Validation Loss*     |         |         |         |         |         |         |         |         |         |         |         |         |
> | RoPE                 | 3.2498  | 3.1297  | 3.0754  | 3.0492  | 3.0112  | 2.9911  | 2.8477  | 2.9711  | 2.9641  | 2.9526  | 2.9503  | 2.9464  |
> | + RoPE++$_\text{EH}$ | 3.2768  | 3.1569  | 3.1021  | 3.0785  | 3.0406  | 3.0167  | 2.8744  | 2.9949  | 2.9904  | 2.9799  | 2.9780  | 2.9681  |
> | + RoPE++$_\text{EC}$ | 3.2420  | 3.1236  | 3.0709  | 3.0466  | 3.0086  | 2.9857  | 2.8456  | 2.9669  | 2.9577  | 2.9515  | 2.9488  | 2.9433  |
> |*Average Score*       |         |         |         |         |         |         |         |         |         |         |         |         |
> | RoPE                 |**40.9** |**41.5** |**42.2** |**42.7** |**43.0** |**43.2** | 42.9    | 42.7    |**43.2** |**43.8** | 43.4    | 43.0    |
> | + RoPE++$_\text{EH}$ | 39.6    | 40.8    | 41.8    | 41.9    | 42.1    | 42.5    |**43.6** |**43.3** | 42.4    | 43.4    | 42.8    | 42.7    |
> | + RoPE++$_\text{EC}$ | 39.6    | 41.2    | 41.4    | 42.0    | 42.7    | 42.4    | 42.9    | 42.7    | 42.7    | 43.0    |**43.8** |**43.1** |

---

### Official Review · Reviewer_aL3q · 2025-11-01

**Soundness:** 2
**Presentation:** 3
**Contribution:** 3
**Rating:** 6
**Confidence:** 3

**Summary:**

This paper proposes RoPE++, an extension of Rotary Position Embeddings (RoPE) that reintroduces the imaginary component of complex-valued attention scores, which standard RoPE discards. The authors argue that this imaginary component contains valuable phase information for long-range dependencies. They propose two variants and the method is evaluated different sized parameter models using short-context and long-context benchmarks .

**Strengths:**

1. Novel perspective: Identifying and addressing the discarded imaginary component in RoPE is creative and theoretically motivated. The observation that imaginary attention captures longer-range dependencies is interesting.
2. Theoretical justification: The paper provides mathematical grounding (Equations 2-5) showing that imaginary attention follows a sine integral characteristic curve, complementing the cosine integral of real attention.
3. Generalization: Method generalizes to diffusion/bidirectional attention models.
3. Dual efficiency benefits: RoPE++_EH achieving comparable performance with half the KV cache is valuable for long-context scenarios.
4. Comprehensive evaluation: Testing across multiple model sizes, benchmarks, and ablations (attention pattern analysis, noise injection experiments) strengthens the empirical validation.

**Weaknesses:**

1. Limited scale: Experiments only go to 700m parameters, which is quite small by modern LLM standards. It's unclear if benefits hold at 7b+ scale where most practical long-context work happens.
2. Modest improvements: Performance gains are often marginal. In Table 2, RoPE++_EC only outperforms RoPE by ~1-2 points on average.
3. No plug-and-play extrapolation: The authors acknowledge (Section 5.3, Limitation) that RoPE++ doesn't provide direct length extrapolation like other methods, limiting practical applicability.
4. Incomplete comparisons: No comparison with recent long-context methods (e.g., YaRN, which is cited but not compared against after long-context training) ALiBi and NoPE trained at 1k context vs. 4k for others makes comparisons less fair.
5. Incomplete ablations: No ablation study of the contribution of the imaginary component. For example, we could use 75% imaginary and 25% real.

**Questions:**

1. Have you tested or do you have plans to validate RoPE++ at 7b or larger scales? Are there theoretical or practical reasons to expect different behavior?
2. Can RoPE++ be combined with other long-context techniques? (YaRN, sparse attention, etc.) Have you explored this?
3. Does RoPE++ converge at similar rates to standard RoPE? Are there any training stability issues? Could you show pretraining loss curves?
4. What happens if we only apply the imaginary part of the RoPE? What would be the effects of doing this?
5. For Figure 5's noise experiment, what are the results with noise added to both components simultaneously?
6. What happens if you use different rotation angles (not -π/2) for the imaginary component?

---

> ### Author Response · Authors · 2025-11-25
> **Reply (1/6)**
>
> First, we sincerely thank the reviewer for recognizing our novel insight, theoretical justification, dual efficiency benefits, and comprehensive evaluation.
>
> **Regarding W1 and Q1 on limited scale**
>
> Thank you for this critical question. Scaling is essential for architecture validation, although concurrent or earlier work still only performs pre-training validation on model sizes smaller than 1B and data volumes smaller than 50B tokens [1-3]. Unfortunately, our available resources limit us to scales below 7B. After an extended effort, we have completed a 1.5B model trained on 50B tokens in 4k context length, followed by 5B tokens in 32k context length, using the same training hyper-parameters used for 776M/376M. The results below demonstrate that RoPE++ still outperforms RoPE. To sum up, within the limits of our available computing resources, we have successfully validated the effectiveness of RoPE++ on all three model scales. We will continue training the 1.5B model and report results after further tokens in the revised paper. We appreciate the reviewer’s inquiry.
>
> |                      | Wiki  | LMD   | TQA   | PIQA  | Hella | Wino  | ARC-e | GPQA  | SIQA  | OBQA  | SG    | Avg   |
> |:---------------------|:-----:|:-----:|:-----:|:-----:|:-----:|:-----:|:-----:|:-----:|:-----:|:-----:|:-----:|:-----:|
> | ***1.5B Short***     | ppl   | ppl   | acc   | acc   | acc   | acc   | acc   | acc   | acc   | acc   | acc   |       |
> | RoPE                 | 27.9  |**15.3**| 36.4  | 70.8  | 46.2  | 53.2  | 44.8  | 25.8  | 39.4  | 24.6  | 44.8  | 42.9  |
> | + RoPE++$\_\text{EH}$ | 28.0  | 16.0  | 37.1  | 71.2  | 45.6  | 53.6  | 45.9  | 25.8  | 40.8  | 27.8  | 44.3  |**43.6**|
> | + RoPE++$\_\text{EC}$ |**27.4**| 15.5  | 35.1  | 69.5  | 46.3  | 53.3  | 43.6  | 28.3  | 40.5  | 23.6  | 45.6  | 42.9  |
>
> |                      | RU    | LE    | R     |       |       |       | BA    | BI    | Lo    | ng    |       |       |       |
> |:---------------------|:-----:|:-----:|:-----:|:-----:|:-----:|:-----:|:-----:|:-----:|:-----:|:-----:|:-----:|:-----:|:-----:|
> | ***1.5B Long***      | 4k    | 8k    | 16k   | 32k   | 64k   | Avg   | 2k    | 4k    | 8k    | 16k   | 32k   | 64k   | Avg   |
> | RoPE                 | 50.5  | 39.9  | 34.6  |**31.9**| 18.3  | 35.1  | 25.9  | 29.7  |**31.8**|**30.5**|**18.5**| 13.5  | 29.5  |
> | + RoPE++$\_\text{EH}$ | 42.5  | 38.1  | 33.1  | 28.7  | 12.7  | 31.0  |**40.0**|**35.9**| 30.5  | 25.1  | 14.7  |**15.8**|**32.9**|
> | + RoPE++$\_\text{EC}$ |**53.1**|**45.7**|**39.0**| 30.6  |**18.9**|**37.5**| 25.1  | 28.3  | 21.4  | 16.6  | 12.6  | 12.3  | 22.9  |
>
> **Regarding W2 on modest improvements**
>
> Thank you for your question. Our modification of RoPE is theoretically grounded. The standard implementation discards the imaginary component of the complex product, and we re-incorporate it into attention. As you have mentioned, Equations 2–5 show that imaginary attention follows a sine-integral curve that complements the cosine integral of real attention. After re-incorporation, we obtain consistent downstream gains. Table 1 shows a consistent advantage over the original RoPE on the average score of multiple short-text tasks. Table 2 shows the same lead on long-context tasks. At 776M, although RoPE++$\_\text{EC}$ surpasses RoPE by only 1-2 points on average, yet as stated in Section 4.3, RoPE++ (both EC and EH variants) delivers more stable performance as context length increases and consistently achieves the best 64k extrapolation, while RoPE++$\_\text{EH}$ matches vanilla RoPE with half the KV-cache and QKV parameters. These results confirm that our approach remains promising.
>
> **Regarding W3 on no plug-and-play extrapolation**
>
> Thank you for your question. Although RoPE++ does not offer plug-and-play extrapolation, this does not diminish its key contribution. We identify that RoPE discards the imaginary component of its complex calculation and show that re-incorporating this component improves downstream performance even when halving cache and attention-head parameters. From this perspective, plug-and-play extrapolation is not a prerequisite property for RoPE++ and its related work. The concurrent HoPE[1], the earlier Partial RoPE adopted by Pythia[4] and MLA[5], and the even earlier ALiBi[6] (only able to extrapolate under the measurement of perplexity) all likewise fail to provide it. Indeed, RoPE itself is an evolution of sinusoidal position embedding and is also not plug-and-play for extrapolation.

---

> ### Author Response · Authors · 2025-11-25
> **Reply (2/6)**
>
> **Regarding W4 on incomplete comparisons**
>
> This is an important question. We provide the results of combining RoPE++ with YaRN in the reply to Q2, and present the updated experiments for a fairer comparison here. We agree that a consistent training context length across all compared methods is crucial for a fair and rigorous evaluation. To address the issue you raised, we have conducted a new set of experiments:
> 1. **ALiBi**: We have addressed an implementation issue in the previous ALiBi baseline by aligning our code with the established open-source model, Baichuan2-13B[7], a model that also uses ALiBi at a 4k context length. The corrected model was then successfully re-trained with a 4k context length. The new training run uses a configuration identical to our proposed method and other baselines, ensuring a fair and correct comparison.
> 2. **NoPE**: Our initial choice of a 1k context for the NoPE baseline was due to significant performance degradation we observed when attempting to train it directly at a 4k context length. This finding aligns with the fact that, to our knowledge, there are no established reports of mainstream fully attention-based LLMs being successfully trained from scratch using NoPE at such context lengths. In an effort to address concerns about experimental fairness and to ensure our comparisons are grounded in a reliable setup, we have taken the reviewer ADoG's suggestion to adopt a reproducible pre-training recipe. Therefore, to provide a stronger and more relevant comparison at the 4k context length, we replaced NoPE with Partial RoPE (as used in Pythia[4]). We then trained this more robust baseline at 4k, ensuring it follows the same configuration as our proposed method.
>
> The updated evaluation results of the 376M and 776M models under this fair comparison framework are presented in the table below.
>
> | | Wiki | LMD | TQA | PIQA | Hella | Wino | ARC-e | GPQA | SIQA | OBQA | SG | Avg |
> |:-|:-:|:-:|:-:|:-:|:-:|:-:|:-:|:-:|:-:|:-:|:-:|:-:|
> | | ppl | ppl | acc | acc | acc | acc | acc | acc | acc | acc | acc | |
> | ***776M Short*** | | | | | | | | | | | | |
> | RoPE | 14.8 | 27.3 | 35.5 | 70.1 | 43.7 | 52.3 | 43.4 | 25.8 | 41.3 | 21.8 | 43.6 | 42.0 |
> | RoPE++$_\text{EH}$ | 15.6 | 28.1 | 35.4 | 69.6 | 42.7 | 53.5 | 45.0 | 25.8 | 41.6 | 26.8 | 42.4 | 42.5 |
> | RoPE++$_\text{EC}$ | 14.8 | 27.3 | 36.1 | 69.3 | 43.6 | 52.3 | 43.7 | 28.3 | 40.1 | 27.6 | 44.4 |**42.8**|
> | FoPE | **14.7** | 27.1 | 33.6 | 68.7 | 43.4 | 52.9 | 45.0 | 24.8 | 39.7 | 24.8 | 45.4 | 42.0 |
> | Pythia | 14.8 |**26.9**| 35.8 | 68.8 | 42.9 | 52.1 | 39.5 | 22.2 | 42.0 | 21.2 | 43.6 | 40.9 |
> | ALiBi | 15.2 | 28.3 | 35.2 | 70.2 | 43.7 | 53.6 | 43.2 | 23.7 | 40.6 | 27.6 | 45.9 | 42.6 |
> | ***376M Short*** | | | | | | | | | | | | |
> | RoPE | 19.9  | 32.7 | 35.5 | 66.3 | 34.8 | 50.9 | 39.3 | 24.8 | 38.6 | 27.4 | 43.7 | 40.1 |
> | RoPE++$_\text{EH}$ | 20.8 | 33.6 | 36.3 | 66.4 | 34.5 | 52.5 | 40.9 | 23.7 | 40.5 | 24.8 | 43.2 | 40.3 |
> | RoPE++$_\text{EC}$ | 19.4 |**32.6**| 37.3 | 68.0 | 35.6 | 53.0 | 41.3 | 25.8 | 40.3 | 23.2 | 44.8 |**41.0**|
> | FoPE |**19.3**| 33.0 | 33.8 | 65.9 | 34.5 | 53.0 | 37.0 | 28.8 | 39.5 | 24.2 | 43.6 | 40.0 |
> | Pythia | 19.2  | 32.9  | 34.7  | 65.8  | 34.9  | 51.5  | 41.3  | 21.2  | 39.7  | 25.6  | 42.5  | 39.7  |
> | ALiBi | 21.2  | 34.6  | 33.8  | 66.1  | 34.2  | 51.1  | 44.4  | 24.8  | 38.7  | 27.4  | 43.9  | 40.5  |
>
> As the revised results demonstrate, our RoPE++$_\text{EC}$ continues to hold a performance advantage over ALiBi and the Partial RoPE baselines on downstream tasks. These results provide stronger evidence for our central claim, confirming our method's effectiveness when evaluated against correctly implemented and fairly trained baselines. Thank you again for your valuable suggestion, which has improved the quality of our work.
>
> **Regarding W5, Q4, and Q6 on the experiment on different ratios or forms of imaginary attention**
>
> Thank you for raising this important question. We need first to clarify that in Section 3.3, from Line 255 to Line 259, we have stated that imaginary and real attention must share the same parameters $W_q,W_k,W_v$, though they are computed independently and treated as separate heads. Allocating distinct subsets of heads to imaginary and real attention would effectively collapse back to standard RoPE, since rotating $q_t$ in imaginary attention by $\pi/2$ yields real attention, with no architecture modification. In other words, imaginary attention is defined relative to real attention and cannot exist independently. Therefore, configurations such as 75% imaginary vs. 25% real or 100% imaginary (applying only the imaginary part) are impossible under RoPE++. Besides, imaginary attention is exactly the product between $k_s$ and  $q_t$ rotated with $-\pi/2$. The results under other rotations are not imaginary attention. We will revise the text to emphasize this point and ensure readers readily grasp the constraint.

---

> ### Author Response · Authors · 2025-11-25
> **Reply (3/6)**
>
> **Regarding Q2 on combining with other long-context techniques**
>
> Thank you for this constructive review. RoPE++ can not only be combined with NTK for context extension during long-context training, but can also be combined with Linear PI[8] and YaRN[9]. Across 376M and 776M model sizes, we conduct extensive experiments, with the interpolation coefficient $s = 8$ for Linear PI and $s = 32$ for YaRN. RoPE++ consistently achieves the highest composite scores on RULER, BABILong, and the average score of short-context tasks, confirming its advantage and generalization. These results will be added to the revised paper.
>
> |                      | RU    | LE    | R     |       |       | BA    | BI    | Lo    | ng    |       |       |
> |:---------------------|:-----:|:-----:|:-----:|:-----:|:-----:|:-----:|:-----:|:-----:|:-----:|:-----:|:-----:|
> |                      | 4k    | 8k    | 16k   | 32k   | Avg   | 2k    | 4k    | 8k    | 16k   | 32k   | Avg   |
> |***776M Long***       |       |       |       |       |       |       |       |       |       |       |       |
> | RoPE + NTK           | 37.4  | 35.1  | 33.0  | 21.2  | 31.7  | 33.5  | 30.7  | 23.6  | 22.0  | 15.1  | 25.0  |
> | + RoPE++$_\text{EH}$ | 38.7  | 35.4  | 33.8  | 24.6  | 33.1  | 31.9  | 26.5  | 18.6  | 16.2  | 11.0  | 20.8  |
> | + RoPE++$_\text{EC}$ | 42.7  | 38.6  | 33.4  | 21.7  |**34.1**| 32.4  | 29.9  | 24.4  | 24.5  | 18.6  |**26.0**|
> | RoPE + PI            | 37.8  | 34.4  | 30.5  | 13.4  | 29.0  | 15.3  | 16.9  | 12.7  | 11.8  | 9.3   | 13.2  |
> | + RoPE++$_\text{EH}$ | 37.9  | 35.0  | 27.5  | 14.6  | 28.8  | 21.0  | 22.4  | 17.1  | 13.7  | 11.1  |**17.1**|
> | + RoPE++$_\text{EC}$ | 43.0  | 38.7  | 28.8  | 13.6  |**31.0**| 25.7  | 23.4  | 16.4  | 9.4   | 8.0   | 16.6  |
> | RoPE + YaRN          | 37.6  | 35.0  | 33.9  | 27.5  | 33.5  | 26.9  | 25.6  | 19.5  | 16.4  | 12.2  | 20.1  |
> | + RoPE++$_\text{EH}$ | 37.9  | 34.9  | 32.2  | 26.1  | 32.8  | 28.0  | 23.9  | 18.6  | 17.8  | 11.7  | 20.0  |
> | + RoPE++$_\text{EC}$ | 42.9  | 36.5  | 36.3  | 22.2  |**34.4**| 26.3  | 24.1  | 21.1  | 19.8  | 16.9  |**21.6**|
> |***376M Long***       |       |       |       |       |       |       |       |       |       |       |       |
> | RoPE + NTK           | 31.6  | 25.6  | 22.0  | 9.5   | 22.2  | 17.7  | 16.1  | 9.1   | 9.4   | 5.9   | 11.6  |
> | + RoPE++$_\text{EH}$ | 29.9  | 28.4  | 17.6  | 9.4   | 21.3  | 14.1  | 15.6  | 12.2  | 9.9   | 8.3   | 12.0  |
> | + RoPE++$_\text{EC}$ | 36.1  | 33.0  | 29.1  | 17.7  |**29.0**| 19.8  | 19.8  | 16.1  | 15.8  | 12.3  |**16.8**|
> | RoPE + PI            | 36.5  | 33.6  | 19.7  | 10.6  | 25.1  | 19.3  | 12.3  | 10.2  | 10.9  | 10.9  | 12.7  |
> | + RoPE++$_\text{EH}$ | 28.0  | 27.6  | 15.8  | 6.9   | 19.6  | 13.3  | 12.4  | 12.8  | 8.9   | 10.4  | 11.6  |
> | + RoPE++$_\text{EC}$ | 37.0  | 32.4  | 28.3  | 10.6  |**27.1**| 24.0  | 20.7  | 15.9  | 14.3  | 12.3  |**17.4**|
> | RoPE + YaRN          | 36.4  | 32.9  | 28.4  | 15.0  | 28.2  | 22.4  | 16.4  | 11.4  | 10.7  | 11.1  | 14.4  |
> | + RoPE++$_\text{EH}$ | 32.7  | 30.2  | 24.9  | 10.7  | 24.7  | 8.7   | 9.3   | 12.1  | 11.3  | 10.9  | 10.5  |
> | + RoPE++$_\text{EC}$ | 36.0  | 33.9  | 31.7  | 17.8  |**29.8**| 27.4  | 23.6  | 18.0  | 16.9  | 12.3  |**19.6**|

---

> ### Author Response · Authors · 2025-11-25
> **Reply (4/6)**
>
> |                      | Wiki  | LMD   | TQA   | PIQA  | Hella | Wino  | ARC-e | GPQA  | SIQA  | OBQA  | SG    | Avg   |
> |:---------------------|:-----:|:-----:|:-----:|:-----:|:-----:|:-----:|:-----:|:-----:|:-----:|:-----:|:-----:|:-----:|
> |                      | ppl   | ppl   | acc   | acc   | acc   | acc   | acc   | acc   | acc   | acc   | acc   |       |
> |***776M Long***       |       |       |       |       |       |       |       |       |       |       |       |
> | RoPE + NTK           | 14.6  | 27.3  | 35.1  | 68.9  | 43.1  | 51.5  | 47.6  | 21.7  | 40.7  | 20.2  | 42.6  | 41.3  |
> | + RoPE++$_\text{EH}$ | 15.3  | 28.1  | 35.4  | 69.9  | 41.9  | 52.6  | 43.2  | 28.3  | 41.0  | 22.2  | 43.4  | 42.0  |
> | + RoPE++$_\text{EC}$ |**14.4**|**27.1**| 35.2  | 70.4  | 43.7  | 52.6  | 44.8  | 31.8  | 40.8  | 27.6  | 44.3  |**43.5**|
> | RoPE + PI            |**27.8**|**14.8**| 35.8  | 65.4  | 33.8  | 53.1  | 38.8  | 25.3  | 39.6  | 27.2  | 44.7  | 40.4  |
> | + RoPE++$_\text{EH}$ | 28.8  | 15.6  | 37.0  | 66.0  | 33.1  | 52.6  | 40.2  | 25.8  | 38.1  | 26.4  | 44.0  | 40.4  |
> | + RoPE++$_\text{EC}$ |**27.8**| 14.9  | 37.0  | 65.4  | 33.9  | 51.7  | 39.5  | 23.2  | 39.4  | 28.4  | 45.8  |**40.5**|
> | RoPE + YaRN          |**27.3**|**14.5**| 36.7  | 65.6  | 33.9  | 51.1  | 37.7  | 29.3  | 39.7  | 27.6  | 46.3  | 40.9  |
> | + RoPE++$_\text{EH}$ | 28.3  | 15.1  | 38.0  | 66.1  | 33.6  | 51.6  | 40.9  | 26.3  | 39.3  | 26.0  | 43.9  | 40.6  |
> | + RoPE++$_\text{EC}$ |**27.3**|**14.5**| 37.1  | 65.8  | 34.8  | 53.4  | 41.8  | 27.8  | 40.1  | 27.2  | 45.1  |**41.5**|
> |***376M Long***       |       |       |       |       |       |       |       |       |       |       |       |
> | RoPE + NTK           | 20.4  |**33.8**| 35.4  | 64.9  | 34.1  | 50.6  | 40.4  | 21.2  | 39.4  | 27.4  | 43.5  | 39.6  |
> | + RoPE++$_\text{EH}$ | 21.7  | 34.8  | 35.2  | 64.5  | 34.3  | 49.9  | 41.5  | 22.7  | 40.0  | 27.0  | 43.1  | 39.8  |
> | + RoPE++$_\text{EC}$ |**20.0**| 33.9  | 37.1  | 66.1  | 34.1  | 53.4  | 38.1  | 21.2  | 39.2  | 28.4  | 43.7  |**40.1**|
> | RoPE + PI            |**33.4**| 20.1  | 34.9  | 69.4  | 43.1  | 52.5  | 42.9  | 29.3  | 40.0  | 21.6  | 44.3  | 42.0  |
> | + RoPE++$_\text{EH}$ | 34.7  | 20.7  | 35.5  | 68.4  | 41.1  | 53.7  | 44.6  | 22.7  | 40.0  | 26.0  | 42.9  | 41.7  |
> | + RoPE++$_\text{EC}$ | 33.7  |**19.4**| 36.6  | 70.1  | 43.1  | 51.9  | 44.8  | 27.3  | 40.7  | 27.0  | 44.1  |**42.8**|
> | RoPE + YaRN          |**32.8**| 19.3  | 36.1  | 69.9  | 43.5  | 52.6  | 44.1  | 25.3  | 41.2  | 24.0  | 43.4  | 42.2  |
> | + RoPE++$_\text{EH}$ | 33.9  | 20.1  | 35.7  | 68.8  | 41.8  | 55.0  | 45.2  | 25.8  | 40.4  | 24.6  | 42.2  | 42.2  |
> | + RoPE++$_\text{EC}$ | 32.9  |**18.8**| 36.7  | 69.5  | 43.6  | 52.6  | 44.1  | 29.3  | 41.0  | 30.2  | 44.0  |**43.4**|
>
> **Regarding Q3 on training details**
>
> Thank you for this important question. We have tracked training loss, validation loss, and the average score of short-context tasks for our 376M, 776M, and 1.5B checkpoints at 5B, 10B, 15B, 20B, 30B, 40B, and 50B tokens. RoPE++ exhibits loss curves that almost overlap with those of RoPE, showing no training stability issues. Although RoPE may show an advantage in the early checkpoints, RoPE++ continues to improve and ultimately surpasses RoPE in average scores. These results and the complete training-loss curves will be added to the revised paper.
>
> | Training Loss        | 5B      | 10B     | 15B     | 20B     | 30B     | 40B     | 50B     |
> |:---------------------|:-------:|:-------:|:-------:|:-------:|:-------:|:-------:|:-------:|
> |***1.5B***            |         |         |         |         |         |         |         |
> | RoPE                 | 3.1798  | 3.0462  | 2.9492  | 2.9048  | 2.9835  | 2.8967  | 2.9447  |
> | + RoPE++$_\text{EH}$ | 3.2051  | 3.0718  | 2.9724  | 2.9333  | 3.0104  | 2.9244  | 2.9688  |
> | + RoPE++$_\text{EC}$ | 3.1708  | 3.0384  | 2.9418  | 2.9025  | 2.9824  | 2.8916  | 2.9404  |
> |***776M***            |         |         |         |         |         |         |         |
> | RoPE                 | 3.2826  | 3.1621  | 3.0279  | 3.1418  | 3.0889  | 3.0913  | 2.9483  |
> | + RoPE++$_\text{EH}$ | 3.3104  | 3.1955  | 3.0622  | 3.1701  | 3.1236  | 3.1237  | 2.9809  |
> | + RoPE++$_\text{EC}$ | 3.2797  | 3.1620  | 3.0294  | 3.1377  | 3.0890  | 3.0889  | 2.9485  |
> |***376M***            |         |         |         |         |         |         |         |
> | RoPE                 | 3.4698  | 3.3576  | 3.2249  | 3.3459  | 3.2945  | 3.2946  | 3.1665  |
> | + RoPE++$_\text{EH}$ | 3.4904  | 3.3821  | 3.2511  | 3.3708  | 3.3202  | 3.3194  | 3.1931  |
> | + RoPE++$_\text{EC}$ | 3.4567  | 3.3473  | 3.2203  | 3.3373  | 3.2848  | 3.2904  | 3.1612  |

---

> ### Author Response · Authors · 2025-11-25
> **Reply (5/6)**
>
> | Validation Loss      | 5B      | 10B     | 15B     | 20B     | 30B     | 40B     | 50B     |
> |:---------------------|:-------:|:-------:|:-------:|:-------:|:-------:|:-------:|:-------:|
> |***1.5B***            |         |         |         |         |         |         |         |
> | RoPE                 | 3.2498  | 3.1297  | 3.0754  | 3.0492  | 3.0112  | 2.9911  | 2.8477  |
> | + RoPE++$_\text{EH}$ | 3.2768  | 3.1569  | 3.1021  | 3.0785  | 3.0406  | 3.0167  | 2.8744  |
> | + RoPE++$_\text{EC}$ | 3.2420  | 3.1236  | 3.0709  | 3.0466  | 3.0086  | 2.9857  | 2.8456  |
> |***776M***            |         |         |         |         |         |         |         |
> | RoPE                 | 3.3480  | 3.2279  | 3.1842  | 3.1552  | 3.1214  | 3.1083  | 2.9661  |
> | + RoPE++$_\text{EH}$ | 3.3865  | 3.2648  | 3.2184  | 3.1877  | 3.1545  | 3.1401  | 2.9988  |
> | + RoPE++$_\text{EC}$ | 3.3541  | 3.2305  | 3.1872  | 3.1550  | 3.1205  | 3.1061  | 2.9664  |
> |***376M***            |         |         |         |         |         |         |         |
> | RoPE                 | 3.5509  | 3.4358  | 3.3933  | 3.3629  | 3.3299  | 3.3177  | 3.1881  |
> | + RoPE++$_\text{EH}$ | 3.5772  | 3.4618  | 3.4217  | 3.3907  | 3.3567  | 3.3430  | 3.2141  |
> | + RoPE++$_\text{EC}$ | 3.5362  | 3.4254  | 3.3870  | 3.3569  | 3.3251  | 3.3140  | 3.2683  |
>
> | Average Score        | 5B    | 10B   | 15B   | 20B   | 30B   | 40B   | 50B   |
> |:---------------------|:-----:|:-----:|:-----:|:-----:|:-----:|:-----:|:-----:|
> |***1.5B***            |       |       |       |       |       |       |       |
> | RoPE                 |**40.9**|**41.5**|**42.2**|**42.7**|**43.0**|**43.2**| 42.9  |
> | + RoPE++$_\text{EH}$ | 39.6  | 40.8  | 41.8  | 41.9  | 42.1  | 42.5  |**43.6**|
> | + RoPE++$_\text{EC}$ | 39.6  | 41.2  | 41.4  | 42.0  | 42.7  | 42.4  | 42.9  |
> |***776M***            |       |       |       |       |       |       |       |
> | RoPE                 | 39.3  | 40.8  |**41.3**|**41.5**| 41.4  | 41.5  | 42.0  |
> | + RoPE++$_\text{EH}$ |**40.0**|**40.9**| 41.2  | 40.6  | 41.7  |**42.2**| 42.5  |
> | + RoPE++$_\text{EC}$ | 38.7  | 40.4  | 41.1  | 41.0  |**41.8**| 41.7  |**42.8**|
> |***376M***            |       |       |       |       |       |       |       |
> | RoPE                 |**39.2**|**39.6**|**40.4**|**39.9**| 39.5  | 39.8  | 40.1  |
> | + RoPE++$_\text{EH}$ | 38.2  |**39.6**| 40.3  | 38.9  | 39.5  | 39.8  | 40.3  |
> | + RoPE++$_\text{EC}$ | 38.8  | 38.8  | 39.3  | 39.6  |**40.3**|**40.2**|**41.0**|
>
> **Regarding Q5 on noise experiment**
>
> This is an interesting question. We have conducted experiments in Section 5.2. We inject Gaussian noise with equal standard deviation to the imaginary and real attention components separately and compare the resulting change in RoPE++ performance on average RULER-4k score. The results demonstrate that noise in the imaginary part degrades long-context performance more severely, as shown in Figures 5e and 5j. In response to the reviewer’s suggestion, we have added experiments where noise is applied to both components simultaneously. The updated results are listed in the table below. Under the same standard deviation $\sigma$, perturbing both parts affects twice as many dimensions, so the joint-noise condition produces a markedly larger drop than noise applied to either component alone.
>
> | $\sigma$                | 0.0   | 0.2   | 0.4   | 0.6   | 0.8   | 1.0   | 1.2   | 1.5  |
> |:------------------------|:-----:|:-----:|:-----:|:-----:|:-----:|:-----:|:-----:|:----:|
> |376M RoPE++$_\text{EC}$  |       |       |       |       |       |       |       |      |
> | Real                    | 31.6  | 31.6  | 31.1  | 29.5  | 25.2  | 17.6  | 10.2  | 3.8  |
> | Imag                    | 31.6  | 31.1  | 30.0  | 26.2  | 19.4  | 12.8  | 7.4   | 3.3  |
> | Both                    | 31.6  | 30.4  | 28.2  | 20.7  | 12.0  | 4.2   | 2.4   | 2.8  |
> | 776M RoPE++$_\text{EC}$ |       |       |       |       |       |       |       |      |
> | Real                    | 36.6  | 36.6  | 36.9  | 34.5  | 31.8  | 25.4  | 15.7  | 2.1  |
> | Imag                    | 36.6  | 36.5  | 35.7  | 33.9  | 28.8  | 17.3  | 7.6   | 1.1  |
> | Both                    | 36.6  | 36.7  | 34.8  | 31.1  | 21.5  | 4.5   | 1.1   | 2.5  |

---

> ### Author Response · Authors · 2025-11-25
> **Reply (6/6)**
>
> [1] HoPE: Hyperbolic Rotary Positional Encoding for Stable Long-Range Dependency Modeling in Large Language Models https://www.arxiv.org/abs/2509.05218
>
> [2] PaTH Attention: Position Encoding via Accumulating Householder Transformations https://arxiv.org/abs/2505.16381
>
> [3] Fourier Position Embedding: Enhancing Attention's Periodic Extension for Length Generalization https://arxiv.org/abs/2412.17739
>
> [4] DeepSeek-V2: A Strong, Economical, and Efficient Mixture-of-Experts Language Model https://arxiv.org/abs/2405.04434
>
> [5] Pythia: A Suite for Analyzing Large Language Models Across Training and Scaling https://github.com/EleutherAI/pythia
>
> [6] Train Short, Test Long: Attention with Linear Biases Enables Input Length Extrapolation https://arxiv.org/abs/2108.12409
>
> [7] Baichuan 2: Open Large-scale Language Models https://arxiv.org/abs/2309.10305 https://huggingface.co/baichuan-inc/Baichuan2-13B-Chat
>
> [8] Extending Context Window of Large Language Models via Positional Interpolation https://arxiv.org/abs/2306.15595
>
> [9] YaRN: Efficient Context Window Extension of Large Language Models https://arxiv.org/abs/2309.00071

---

### Author Response · Authors · 2025-12-02
**Summary to AC (1/2)**

Dear AC,

We appreciate your time and effort in reviewing our submission, especially considering the recent challenges. We thank you for your attention to our work.

We also appreciate the reviewers for their valuable and constructive feedback. We are glad that reviewers acknowledge the contribution of our work in recognizing
- novel perspective on identifying the discarded imaginary component in RoPE and incorporating this to capture long-context dependencies, which shows promise, especially for long context (aL3q, ADoG, RADi),
- theoretical explanations to provide mathematical grounding showing that imaginary attention follows a sine integral characteristic curve, complementing the cosine integral of real attention (aL3q, ADoG),
- Dual efficiency benefits of RoPE++$_\text{EH}$ that achieves comparable performance with half the KV cache and QKV parameter, which is valuable for long-context scenarios (aL3q, mcxi), and
- Comprehensive evaluation with different model sizes, extensive datasets, and several baselines, as well as multiple ablations, validating its effectiveness (aL3q, ADoG, RADi, mcxi).

The reviewers also raise a number of questions, which we reply to in the reviewer-specific comments below. Altogether, our responses provide further support for the effectiveness of our RoPE++, an imaginary extension of Rotary Position Embeddings for long-context LLMs.

**Regarding incomplete comparisons** (aL3q, ADoG)

This is a critical question. During rebuttal, we correct the previously reported ALiBi results and, following Reviewer ADoG’s suggestion, replace the NoPE with Partial RoPE. We pre-train 376M and 776M models for 50B tokens in 4k context under identical hyperparameters as RoPE/RoPE++. The detailed results are provided in the reviewer-specific comments below. RoPE++ again surpasses all other RoPE variants, and we will update the paper with these revised experiments.

**Regarding combination with other long-context techniques** (aL3q, RADi, mcxi)

Many reviewers have asked about this point. The original paper only uses NTK to extrapolate both RoPE and RoPE++. In the rebuttal, we additionally evaluate the most frequently mentioned and widely adopted schemes, YaRN and linear interpolation, on 376M and 776M models. Under these settings, RoPE++ still achieves the best average scores on RULER, BABILong, and short-context tasks, confirming its broad generalizability. We will incorporate these new results into the revised paper.

---

> ### Author Response · Authors · 2025-12-02
> **Summary to AC (2/2)**
>
> **Regarding limited scale** (aL3q, ADoG)
>
> We have pre-trained 376M and 776M models for 50B tokens on 4k short texts, followed by long-context training for 5B tokens. Resource constraints limit our model size to 1.5 B. We have trained the 1.5B models with identical hyperparameters and token numbers. At 1.5B, RoPE++ again achieves the highest average scores on RULER, BABILong, and short-context tasks. Concerning the data scale, additional checkpoints within the first 50B tokens, as well as the extended training checkpoints for 776M and 1.5B, show that convergence is stable before the 50B tokens are reached. Regarding the modest improvements noted by Reviewer aL3q, RoPE++ obtains consistent downstream gains, and both EC and EH variants deliver more stable performance as context length increases and consistently achieve the best 64k extrapolation, while RoPE++$_\text{EH}$ matches vanilla RoPE with half the KV-cache and QKV parameters. These results confirm that our approach remains promising.
>
> **Regarding dependency capture** (ADoG, RADi, mcxi)
>
> Many reviewers have also questioned this point. We first use a mathematical derivation to show that, in expectation, imaginary attention decays more slowly with relative distance and assigns more attention to distant tokens, thus offering greater potential for capturing long-context semantic dependencies. This point is clarified in the appendix and in our response to Reviewer RADi, and it is further corroborated by the noise experiment that Reviewer aL3q endorsed. We will reorganize related sections to improve readability. Regarding the redundancy and conflict raised by Reviewer mcxi, our visualizations reveal different attention tendencies between the imaginary and real attentions, and downstream evaluations confirm the incorporation of the imaginary component. Consequently, the potential redundancy or conflict is outweighed by the complementary benefits. We will also try to provide more fine-grained case studies in our future work.
>
> **Regarding training and architectural details** (aL3q, RADi)
>
> In response to Reviewer RADi, we have added comparisons of training throughput and of the impact of RoPE++ on Wq and Wo parameters. Regarding the additional overhead noted by Reviewer mcxi for RoPE++$_\text{EC}$, long-context inference is IO-bound and dominated by KV-cache memory cost, so the absence of extra cache remains acceptable. We will continue to develop more elegant computation optimizations tailored to RoPE++ to reduce this additional computational cost. Concerning Reviewer aL3q’s query about different rotation angles or ratios for imaginary attention, we emphasize that imaginary attention is defined relative to real attention, cannot exist independently, and is exactly the product between $k_s$ and  $q_t$ rotated by $-\pi/2$. The results under other rotations are not imaginary attention. We will revise the text to highlight this constraint and ensure readers readily understand it. The above clarifications will be incorporated into the paper.
>
> **Regarding plug-and-play enhancement** (aL3q, mcxi)
>
> As several reviewers note, our method cannot be applied plug-and-play and achieve training-free length extrapolation. We point out that many classical position embedding works also share this limitation. Even RoPE itself, for example, is an evolution of sinusoidal position embedding and is also not a plug-and-play enhancement. More importantly, we uncover the long-context potential of imaginary attention, demonstrate that its introduction yields stable gains on long-context tasks, and show that comparable performance is achieved with half the KV-cache and QKV-parameter count under RoPE++$_\text{EH}$ configuration. These findings constitute our primary contribution.

---

### Meta-Review · Area_Chair_z6nz · 2026-01-07

**Summary:**

Reviewers appreciated the novel usage of RoPE's imaginary component for long-context modeling, its theoretical grounding via sine/cosine integrals, and efficiency gains (e.g., halved KV cache in RoPE++_EH). Reviewers' concerns are around limited scale (maximum 1.5B parameters), modest performance gains (1-2% average improvement), and lack of plug-and-play applicability resulted in a marginal accept. The rebuttal from authors provided fair comparisons and evaluated larger models, and thus I lean towards acceptance.

**Reviewer Concerns:**

Addressed: Fairer baselines (retrained ALiBi/NoPE at 4k, replaced with Partial RoPE/Pythia); combinations with long-context methods (YaRN/PI/NTK); training stability/loss curves; architectural details (Wq sharing, throughput).

Outstanding: True large-scale validation (7B+ models with 100B+ tokens); potential redundancy/conflicts in dual attention; short-range fluctuations in imaginary curve; training-free extrapolation lacking.

**Reviewer Scores:**

- Reviewer aL3q: Would raise to 8 (rebuttals fix comparisons/ablations, but scale/modesty linger).
- Reviewer ADoG: Would raise to 6 (new fair setups/debugging address skepticism).
- Reviewer RADi: Would maintain 6 (theory clarified via expectations).
- Reviewer mcxi: Would maintain 8 (efficiency/advantages hold; rebuttals confirm no major conflicts).

---

### Decision · Program_Chairs · 2026-01-26

Accept (Poster)